# Diffusion interface layer controlling the acceptor phase of bilayer near-infrared polymer phototransistors with ultrahigh photosensitivity

Tao Han[1✉], Zejiang Wang[1], Ning Shen[1], Zewen Zhou[1], Xuehua Hou[2], Shufang Ding[1], Chunzhi Jiang[1], Xiaoyi Huang[1], Xiaofeng Zhang [iD] [3✉] & Linlin Liu[2]

The narrow bandgap of near-infrared (NIR) polymers is a major barrier to improving the performance of NIR phototransistors. The existing technique for overcoming this barrier is to construct a bilayer device (channel layer/bulk heterojunction (BHJ) layer). However, acceptor phases of the BHJ dissolve into the channel layer and are randomly distributed by the spin-coating method, resulting in turn-on voltages ($V_o$) and off-state dark currents remaining at a high level. In this work, a diffusion interface layer is formed between the channel layer and BHJ layer after treating the film transfer method (FTM)-based NIR phototransistors with solvent vapor annealing (SVA). The newly formed diffusion interface layer makes it possible to control the acceptor phase distribution. The performance of the FTM-based device improves after SVA. $V_o$ decreases from 26 V to zero, and the dark currents decrease by one order of magnitude. The photosensitivity ($I_{ph}/I_{dark}$) increases from 22 to $1.7 \times 10^7$.

---

[1] Hunan Provincial Key Laboratory of Xiangnan Rare-Precious Metals Compounds Research and Application, School of Physics and Electronic Electrical Engineering, Xiangnan University, Chenzhou 423000, People's Republic of China. [2] Institute of Polymer Optoelectronic Materials and Devices, State Key Laboratory of Luminescent Materials and Devices, South China University of Technology, Guangzhou 510640, People's Republic of China. [3] National Engineering Laboratory for Modern Materials Surface Engineering Technology & The Key Lab of Guangdong for Modern Surface Engineering Technology, Guangdong Institute of New Materials, Guangdong Academy of Sciences, Guangzhou 510650, People's Republic of China. ✉email: than@xnu.edu.cn; zxf200808@126.com

Near-infrared (NIR) polymer photodetectors have attracted extensive attention due to their potential for use in bioimaging, health monitoring, and medical applications[1–3]. In general, as a typical photodetector, phototransistors have advantages over photodiodes. Phototransistors can detect light signals that are much weaker than photodiodes because the light can directly reach the active layer of the phototransistors[4]. At the same voltages, the dark currents of the photodiodes are larger than that of the phototransistor due to the short electrode distance (photodiodes: < 500 nm, phototransistors: > 5 μm)[5,6]. However, the NIR polymer materials used in phototransistors have a narrow bandgap, which always leads to a high exciton binding energy and the difficult dissociation of photogenerated excitons[7]. The number of photogenerated excitons in photodetectors is usually increased by introducing abundant acceptor phases[8]. Nevertheless, a high concentration of acceptor phases in traditional single-layer bulk heterojunction (BHJ) photodetectors is needed, which can easily lead to electron/hole recombination[9]. As a result, phototransistors usually have low mobility, high off-state dark currents, and large turn-on voltages ($V_o$)[9]. Furthermore, the photocurrents and photosensitivity ($I_{ph}/I_{dark}$ ratio) of phototransistors, which are strongly related to the shift in the turn-on voltages before and after illumination, decrease with an increase in $V_o$[10].

It has been reported that the design of bilayer devices (channel transport layer/BHJ layer) could effectively enhance device performance[11]. The channel transport layer ensures the high mobility of the bilayer devices. The BHJ layer promotes the dissociation efficiency of photogenerated excitons (large photocurrents)[12–16]. However, mutual dissolution occurs at the interface layer in the bilayer devices[12,17]. As a result, many randomly distributed acceptor phases re-enter the channel transport layer, and the performance of NIR phototransistors becomes worse[11]. It has been found that the problem of mutual dissolution can be solved by using the poor solvent of the bottom film as a good solvent for the preparation of the top film[18] or by adding crosslinking agents into the channel transport layer while spin-coating the second layer[12]. However, it is difficult to prevent the uneven distribution of acceptor phases. The degree of interfacial mutual dissolution is influenced by factors such as the residence time of the spin-coating solution on the first layer surface, spin-coating speed, and dose of solution during the preparation of the second layer.

The performance of photodetectors can be improved by adjusting the acceptor phases, which act as electron capture and storage units. In our previous work, the gain and response speed of lateral polymer photodetectors were controlled by changing the concentration of the acceptor phases[19]. Lateral devices with low dark currents and fast response speeds were prepared by taking advantage of the energy level gradient between ZnO and the acceptor phases[20]. Given the importance of the acceptors, we also established a model to explain the electron transport mechanism. This model was based on the electrical properties of phototransistors doped with different concentrations of acceptor phases[21]. However, to further improve and control the performance of bilayer NIR photodetectors, it was found that an uneven distribution of acceptor phases caused by mutual interfacial dissolution was a major technical challenge.

The floating film transfer method (FTM) is used to prepare organic photoelectric devices. With this method, highly crystalline self-organized active layers with well-stacked domains are easily formed on the water substrate via the spontaneous spreading of the solution[22–26]. For instance, Kumari et al.[25] obtained air-processed FTM-based organic solar cells with 13.8% efficiency. Sung et al.[26] achieved high mobility FTM-based flexible organic field-effect transistors by controlling the solvent evaporation time. Using an FTM to prepare organic photoelectric devices enables the formation

of noninterference bilayer films and avoids interfacial mutual dissolution. However, the large distance between the acceptor phases and channel layer makes it difficult to control device performance by adjusting the acceptor phases in the BHJ. This issue can be solved by employing solvent vapor annealing (SVA) treatment. The fullerene-based acceptor phases easily diffused during the SVA treatments due to their small volume[27,28]. For example, Kim et al.[27] fabricated high-performance bilayer organic photodiodes by controlling donor/acceptor interdiffusion in heterojunctions sandwiched between donor and acceptor layers.

In this work, noninterference bilayer films were prepared by FTM. Subsequently, the FTM-based films were treated with SVA, where the acceptor phases in the BHJ were driven to the channel transport layer. In this way, a controllable diffusion interface can be obtained. The resultant NIR phototransistors exhibited ultrahigh performance. The experimental results and possible mechanism are reported below.

## Results

The structure of the device consists of Si/SiO$_2$/PDPP3T/Au/PDPP3T:PC$_{61}$BM (Fig. 1a), among which the FTM-based poly(diketopyrrolopyrrole-terthiophene) (PDPP3T) and PDPP3T:PC$_{61}$BM layers were treated with SVA (chloroform solvent). Figure 1b and Supplementary Fig. 1 describe the film transfer process. The dropped solution expanded rapidly on the deionized water surface to form a film. After the film was completely dry, the sample was gently placed on the film surface. The sample was then pressed into water with tweezers but immediately removed (Fig. 1b). Finally, high-quality FTM-based films without corner effects were obtained, and multiple samples were prepared simultaneously. This result suggested that the FTM could be used to prepare large-area films, thereby greatly reducing the production cost of photodetectors.

TEM was used to observe the phase separation in the organic blend films[29]. In the TEM images, PC$_{61}$BM appeared dark due to its high electron density, while PDPP3T appeared bright (Fig. 2a). The TEM images of transferred PDPP3T:PC$_{61}$BM films had darker areas than those of the spin-coated films (Supplementary Fig. 2). This phenomenon suggested that the separation of the donor and acceptor phases in the transferred films was poorer, which may be due to the formation of more aggregated PDPP3T phases in the transferred films, thus imposing less effect on the PC$_{61}$BM phases. Additionally, the poor phase separation in the transferred PDPP3T:PC$_{61}$BM films was also confirmed by atomic force microscopy (AFM), as shown in Fig. 2b. More details will be introduced in the following section. Colberts et al.[23] reported that the photoactive material-based solution would spread on a water substrate due to differences in surface tension (Marangoni flow). The poor phase separation in the transferred PDPP3T:PC$_{61}$BM film could be attributed to the different expansion rates of the donor and acceptor phases during the spontaneous spreading of the solution on the water surface. Despite having a low acceptor phase concentration, the blended films with poor phase separation could already produce an obvious light response in the phototransistors due to the existence of gate electrodes[21]. Therefore, the high crystallization of the PDPP3T phases formed in the transferred films would ensure large hole mobility, further promoting devices with large photocurrents.

The absorption peak of the FTM-based PDPP3T/PDPP3T:PC$_{61}$BM bilayers showed a wide wavelength range (300-1100 nm), as shown in Fig. 2c. The absorption of the bilayers in the NIR band (780-1100 nm) was significantly higher than that of the PDPP3T:PC$_{61}$BM single layer. This increased absorption could provide more photogenerated carriers for devices in the NIR band. In addition, grazing-incidence X-ray diffraction

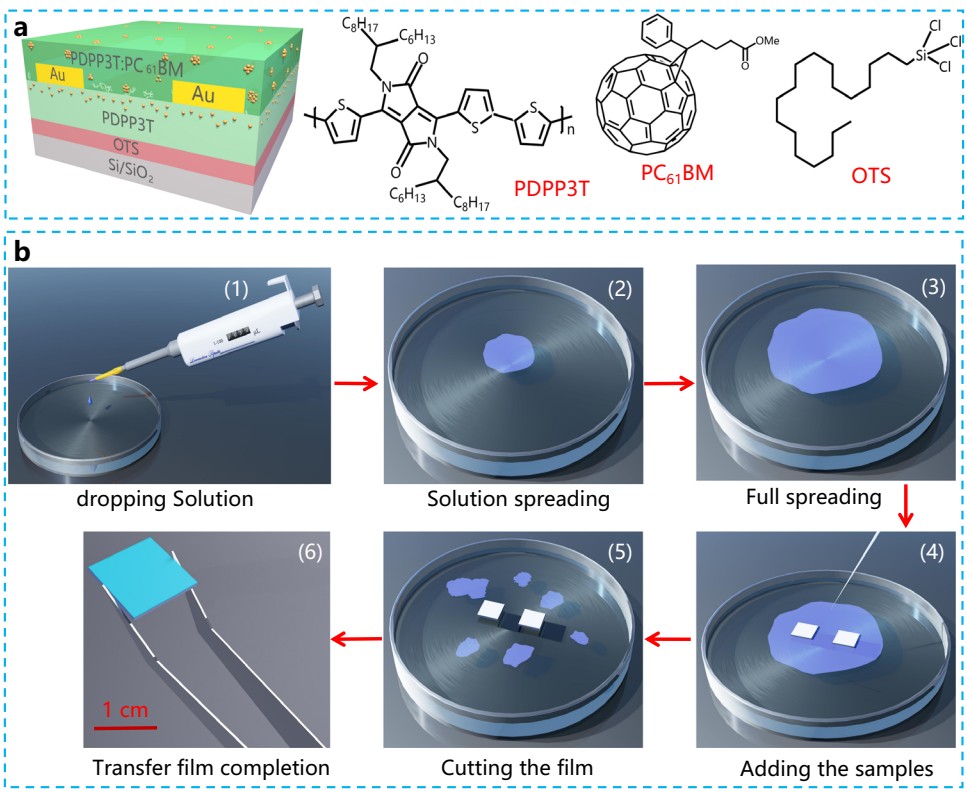

**Fig. 1 Device structure, molecular structures and fabrication process of the FTM-based films. a** Device structure with SVA treatment and the molecular structures of PDPP3T, $PC_{61}BM$ and OTS. **b** Schematic diagram for the fabrication of the FTM-based films.

(GIXRD) of the films showed that the donor phases in the FTM-based PDPP3T:$PC_{61}BM$ films had higher crystallinity (π-π packing) than the spin-coated PDPP3T:$PC_{61}BM$ films (Fig. 2d). Deng et al.[30] reported that the grain size of the organic crystals in the films was remarkably enlarged via a water-surface drag coating method, along with a significant improvement in the π-π packing order. Hence, FTM-based PDPP3T:$PC_{61}BM$ films could be used as the active layers to enhance the transport of carriers by optimizing the π-π stacking. Furthermore, the energy levels of PDPP3T and $PC_{61}BM$ were determined by ultraviolet photoelectron spectroscopy (UPS). The highest occupied molecular orbital (HOMO) levels were obtained from the valence band edges (Fig. 2e) and the secondary electron cutoff (Fig. 2f), indicating that the HOMO levels of the PDPP3T and $PC_{61}BM$ were 5.24 and 5.96 eV, respectively (Supplementary Table 1). The lowest unoccupied molecular orbital (LUMO) levels were determined by combining the optical bandgap from the UV–vis absorption spectra (Fig. 2c and Supplementary Fig. 3) with the HOMO levels. The results show that the LUMO levels of PDPP3T and $PC_{61}BM$ were 3.89 and 4.1 eV, respectively (Supplementary Table 1).

In situ Raman and UV–Vis absorption spectra were obtained to test the acceptor phase diffusion between the two FTM-based films (Fig. 3, Supplementary Figs. 4–5). For the PDPP3T/PDPP3T:$PC_{61}BM$ bilayer, the Raman peak of $PC_{61}BM$ (at 1460 cm$^{-1}$) gradually decreased with increasing SVA temperature (Fig. 3a–b), indicating that the acceptor phases diffused in the films. This observation could be explained by the fact that the volume of $PC_{61}BM$ was smaller than that of the PDPP3T polymer. The $PC_{61}BM$ phases diffused more easily after annealing, and this diffusion followed Fick's second low[31]. The in situ Raman spectra of the PDPP3T/$PC_{61}BM$ bilayer exhibited a wide variety (Supplementary Fig. 4), which was consistent with

the phenomenon observed in the PDPP3T/PDPP3T:$PC_{61}BM$ bilayer. This result also confirmed the existence of the diffusion process.

Film depth-dependent light absorption under low-pressure oxygen plasma etching is commonly used to determine the interface position of films[32]. However, this method may drive the polymer molecules from the top layer into the bottom layer, making it difficult to distinguish weak diffusion interfaces. Hence, in this work, the PEDOT:PSS layer was used to help measure the changes in the absorption peaks of $PC_{61}BM$ (Fig. 3c, d, Supplementary Fig. 5). The films were treated with SVA at various intervals and then immersed in water to remove the PEDOT:PSS/PDPP3T:$PC_{61}BM$ layers. Afterwards, films with quartz/OTS/PDPP3T structures were obtained (Supplementary Fig. 5). In the UV–Vis absorption spectra of those films, the peak intensity of $PC_{61}BM$ increased with prolonged SVA treatment (Fig. 3c, d). This result indicated that SVA treatment promoted the diffusion of $PC_{61}BM$ into the PDPP3T channel transport layer, which also proved the occurrence of acceptor phase diffusion.

Devices with different structures were constructed to investigate the effects of the mutual dissolution interface and diffusion interface on the performance of phototransistors (Fig. 4, Supplementary Figs. 6–9, Tables 1 and 2, Supplementary Tables 2–3). The surface and cross-section morphologies of the bilayer films prepared by different methods are illustrated in Fig. 4d-i. The features of these films are described by the schematic diagram shown in Fig. 4a–c. For the spin-coated bilayer devices, when the CHCl$_3$-soluble $PC_{61}BM$ solution was used, the interface between the PDPP3T and $PC_{61}BM$ films was almost invisible (~184 nm, Fig. 4g), indicating that the two films had a high degree of mutual dissolution. In comparison, when the THF-soluble $PC_{61}BM$ solution was used, a relatively blurred interface between PDPP3T and $PC_{61}BM$ was observed (Fig. 4h), indicating the partial

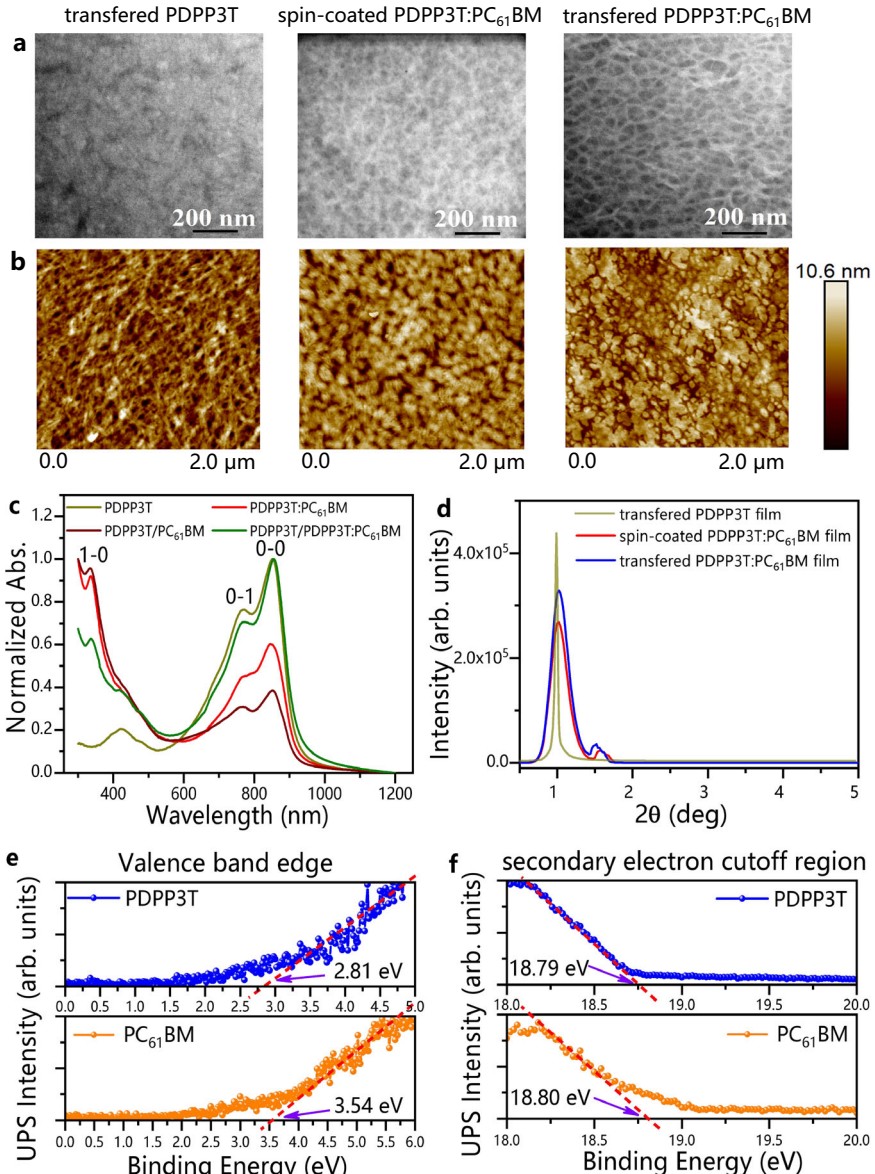

**Fig. 2 Basic parameters of different films. a** TEM. **b** AFM (scanning area is 2 × 2 μm²). **c** UV–vis absorption spectra (0-0 absorption peak at 850 nm, 0-1 absorption peak at 766 nm, 1-0 absorption peak at 334 nm). **d** Grazing-incidence X-ray diffraction (GIXRD) patterns. **e** Ultraviolet photoelectron spectroscopy (UPS) spectra of the valence band region. **f** UPS spectra of the secondary electron cutoff region.

dissolution of these two films. In addition, an obvious interface was observed in the transferred bilayer devices (Fig. 4i), suggesting no mutual dissolution between the first and second layers.

The transferred PDPP3T devices showed a large hysteresis window and weak light response under 850 nm illumination (Supplementary Fig. 8a, b). The bandgap of the PDPP3T material was approximately 1.35 eV (Supplementary Table 1), and abundant electrons injected from the electrode emerged in the active layers, resulting in high turn-on voltages (27 V) and off-state dark currents ($2.0 \times 10^{-5}$ μA, Supplementary Table 3). Compared with PDPP3T devices, the PDPP3T:PC$_{61}$BM devices showed a stronger light response (Supplementary Fig. 8c, d), and turn-on voltages and off-state dark currents decreased simultaneously to some extent (Supplementary Table 3). These drops in the turn-on voltages and dark currents were mainly due to electron capture by the acceptor phases[33,34]. Furthermore, the performance of the

bilayer device spin-coated with good solvent (Device #1: S-PDPP3T/PC$_{61}$BM-CHCl$_3$) was almost identical to that of the PDPP3T:PC$_{61}$BM devices (Table 2, Supplementary Table 3). This was because the PC$_{61}$BM layers spin-coated with CHCl$_3$ could dissolve the bottom PDPP3T layer, forming a large mutually dissolved interface and resulting in abundant acceptor phases entering the channel transport layer. However, the bilayer devices spin-coated with poor solvent (Device #2: S-PDPP3T/PC$_{61}$BM-THF) had a thin mutual dissolution interface layer (Fig. 4b), leading to the acceptor phases having a small effect on the channel transport layer. Therefore, the mobility of Device #2 was comparable to that of the PDPP3T devices (Supplementary Fig. 7, Table 2, Supplementary Table 3). Notably, the acceptor phases could capture electrons injected from the electrodes[33,34], dramatically reducing the turn-on voltages ($V_o \sim 1$ V) of Device #2 (Fig. 4j, l). When the FTM-based bilayer devices (#3 T-PDPP3T/PC$_{61}$BM) were prepared without SVA (Fig. 4c), no mutual

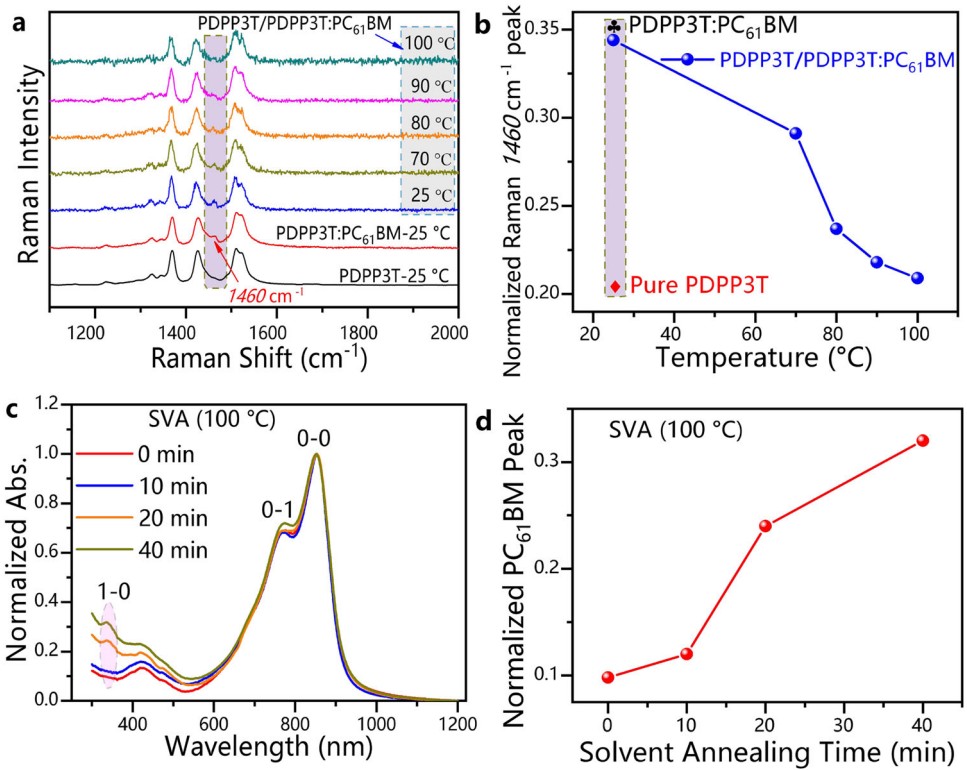

**Fig. 3 The in situ Raman and UV–Vis absorption spectra of the FTM-based films. a** In situ Raman spectra of the PDPP3T/PDPP3T:PC$_{61}$BM bilayers with SVA at different temperatures (PDPP3T-25 °C: the dark line represents the PDPP3T film at 25 °C; PDPP3T:PC$_{61}$BM-25 °C: the red line represents the PDPP3T:PC$_{61}$BM film at 25 °C; PDPP3T/PDPP3T:PC$_{61}$BM bilayers: the blue line represents 25 °C, the brown line represents 70 °C, the orange line represents 80 °C, the magenta line represents 90 °C, and the deep green line represents 100 °C). **b** Dependence of the normalized PC$_{61}$BM Raman peak in the PDPP3T/PDPP3T:PC$_{61}$BM layer on annealing temperature (data from Fig. 3a, ♣ represents the PDPP3T:PC$_{61}$BM film, ♦ represents the PDPP3T film). **c** Absorption spectra of the transferred films after removing the films above the PEDOT:PSS layer with different SVA times (0-0 absorption peak at 850 nm, 0-1 absorption peak at 766 nm, 1-0 absorption peak at 334 nm). **d** Relationship between the normalized PC$_{61}$BM absorption peak and annealing time (data from Fig. 3c).

dissolution interface was formed, so they had electrical properties similar to those of the PDPP3T devices (Table 2, Supplementary Table 3). The effectiveness of SVA treatment on the performance of Device $^{\#}4$ (T-PDPP3T/PC$_{61}$BM-SVA) was further tested. Compared with the PDPP3T devices, Device $^{\#}4$ showed similar hole mobility and lower turn-on voltages ($V_o$ ~ 4 V) and off-state dark currents ($1.8 \times 10^{-6}$ μA), but higher photocurrents (33 μA) and $I_{ph}/I_{dark}$ ($8.3 \times 10^{6}$) (Table 2, Supplementary Table 3, Fig. 4j–l). These results further indicated that the acceptor phases of Device $^{\#}4$ diffused into the channel transport layer after SVA treatment, forming a diffusion interface (Supplementary Fig. 4). Due to the formation of a diffusion interface, the electrical properties of Device $^{\#}4$ were better than those of Device $^{\#}2$, which only had a thin mutually dissolved interface (Fig. 4b).

The FTM-based bilayer devices (PDPP3T/PDPP3T:PC$_{61}$BM) with diffusion interface layers exhibited superior electrical properties in terms of the hole mobility, $\Delta I_{ph}$, responsivity (R), $I_{ph}/I_{dark}$, turn-on voltages, and EQE (Fig. 5, Supplementary Figs. 10–15, Table 3, Supplementary Table 4). The hysteresis window of the FTM-based bilayer SVA devices ($^{\#}6$) decreased significantly, indicating that the defects/traps in the films could be effectively eliminated (Supplementary Figs. 11–12). Hence, the hole mobility of SVA devices ($^{\#}6$) reached 0.752 cm$^2$·V$^{-1}$·s$^{-1}$ (Table 3), which was higher than that of the PDPP3T devices (0.307 cm$^2$·V$^{-1}$·s$^{-1}$, Supplementary Table 3). The $\Delta I_{ph}$ value of the SVA devices ($^{\#}6$) increased over twofold compared to that of the devices without SVA ($V_g = -30$ V, Fig. 5b, Table 3), and the $\Delta I_{ph}$ value continued to increase with light illumination (Fig. 5d). Figure 5c shows that

the R and $I_{ph}/I_{dark}$ of the phototransistors were closely related to the gate voltages. In general, the R value of the phototransistors tended to be large (~ 8700 A/W, Table 3) due to the photogating effect. A device with a high W/L value usually has a high R value. Compared with the W/L value of the device with an ultrahigh R value[35,36], the W/L value of our device was lower. Therefore, the increased R value of the phototransistors prepared in this work might not be related to the high W/L value but could be caused by the increase in the number of photogenerated carriers. In the dark state, the turn-on voltages of the SVA devices ($^{\#}6$) could be reduced to almost zero by controlling the diffusion of the acceptor phases, indicating that this device could be used at low operating voltages. In contrast, under illumination conditions (0.04 mW/cm$^2$, 850 nm), the turn-on voltages of the SVA devices ($^{\#}6$) increased to 27 V, resulting in a large response window (Supplementary Figs. 12–13). This significant turn-on voltage shift between the dark stage and illumination state ensured that the SVA devices ($^{\#}6$) could be adjusted to the accumulation region for illumination and to the depletion region for the dark state[10]. The $I_{ph}/I_{dark}$ value (~$1.7 \times 10^{7}$ @ $V_g = 0$ V, Table 3) of the SVA devices ($^{\#}6$) was significantly higher than that of the device without SVA ($^{\#}5$). This $I_{ph}/I_{dark}$ value was higher than that of the NIR phototransistors reported in the literature[37,38]. The high $I_{ph}/I_{dark}$ value of our phototransistors could be attributed to the negative shift in the turn-on voltages. On the one hand, the dark currents remained at a low level with the aid of the gate electrodes; on the other hand, the photocurrents increased with the negative shift in the turn-on voltages. In addition, the SVA devices ($^{\#}6$) also had high EQE (1000-13000 × 100%) in the wavelength range of

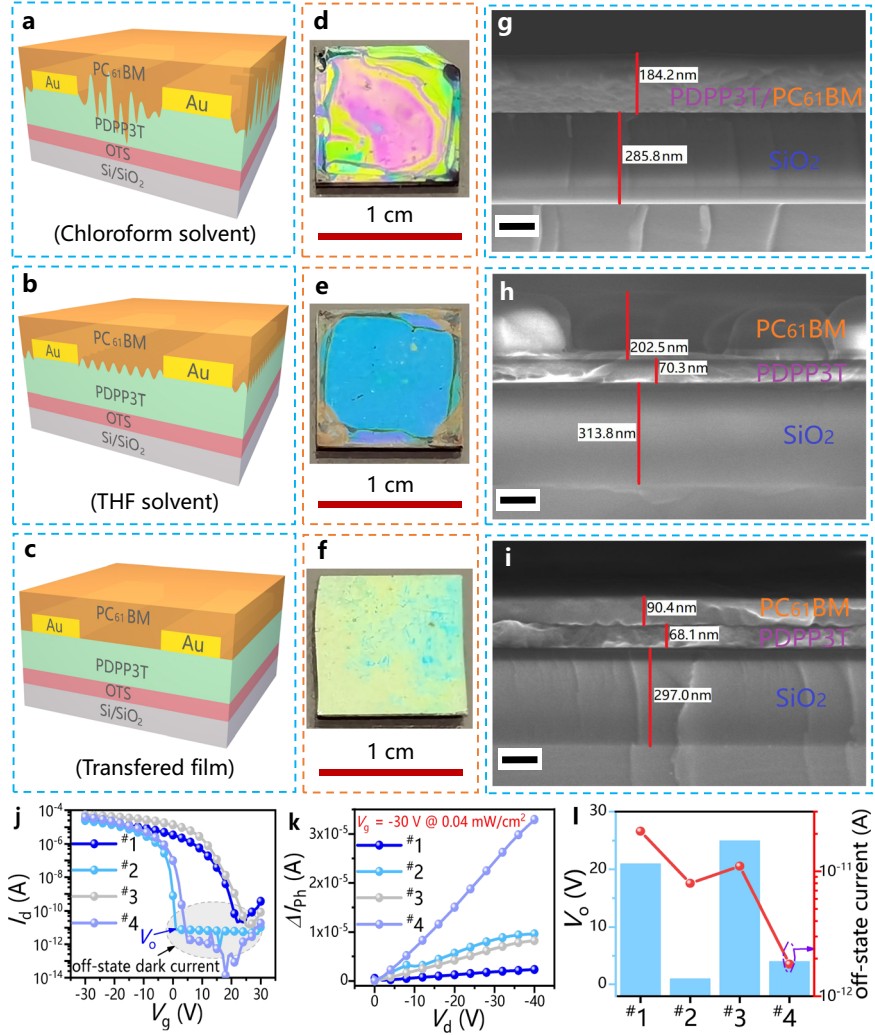

**Fig. 4 Schematic diagram and electrical properties of the Si/SiO$_2$/OTS/PDPP3T/Au/PC$_{61}$BM structured devices via different fabrication methods.**
**a** Schematic diagram of the PC$_{61}$BM solution dissolved in chloroform solvent for the spin-coating bilayer device ($^{#1}$ S-PDPP3T/PC$_{61}$BM-CHCl$_3$).
**b** Schematic diagram of the PC$_{61}$BM solution dissolved in THF solvent for the spin-coating bilayer device ($^{#2}$ S-PDPP3T/PC$_{61}$BM-THF). **c** Schematic diagram of bilayer device via FTM ($^{#3}$ T-PDPP3T/PC$_{61}$BM). **d** ($^{#1}$ S-PDPP3T/PC$_{61}$BM-CHCl$_3$), **e** ($^{#2}$ S-PDPP3T/PC$_{61}$BM-THF), **f** ($^{#3}$ T-PDPP3T/PC$_{61}$BM) Film surface photographs. **g** ($^{#1}$ S-PDPP3T/PC$_{61}$BM-CHCl$_3$), **h** ($^{#2}$ S-PDPP3T/PC$_{61}$BM-THF), **i** ($^{#3}$ T-PDPP3T/PC$_{61}$BM) Cross-sectional SEM images of the films (scale bar is 100 nm). **j** (transfer curve), **k** (dependence of photocurrent ($\Delta I_{ph}$) on the source-drain voltage ($V_d$)), **l** ($V_o$ and off-state dark current) Electrical properties of the devices prepared by different methods. The transfer curves of the devices were measured at a constant $V_d = -30$ V.

**Table 1 Device structures by different preparation methods$^a$.**

| Device | Dielectric layer | First layer | Electrode | Second layer |
|---|---|---|---|---|
| $^{#1}$ S-PDPP3T/PC$_{61}$BM-CHCl$_3$ | SiO$_2$ (300 nm)/OTS | PDPP3T (spin-coated) | Au | PC$_{61}$BM (spin-coated with CHCl$_3$ solvent) |
| $^{#2}$ S-PDPP3T/PC$_{61}$BM-THF | | PDPP3T (spin-coated) | | PC$_{61}$BM (spin-coated with THF solvent) |
| $^{#3}$ T-PDPP3T/PC$_{61}$BM | | PDPP3T (transferred) | | PC$_{61}$BM (transferred and without SVA) |
| $^{#4}$ T-PDPP3T/PC$_{61}$BM-SVA | | PDPP3T (transferred) | | PC$_{61}$BM (transferred and with SVA) |

$^a$ The thicknesses of the PDPP3T film and PDPP3T:PC$_{61}$BM film via the film transfer method are 70 and 60 nm, respectively; the thickness of the PDPP3T film via the spin-coating method is 70 nm.

**Table 2 Performance parameters of the different devices.**

| Device | $\Delta I_{ph}$ (μA) | $V_o$ (V) in the dark | Off-state current (μA) | Mobility (cm$^2$·V$^{-1}$·s$^{-1}$) | $I_{ph}/I_{dark}$ |
|---|---|---|---|---|---|
| $^{#1}$ S-PDPP3T/PC$_{61}$BM-CHCl$_3$ | 2.3 | 21 | $2.1 \times 10^{-5}$ | 0.118 | 4 |
| $^{#2}$ S-PDPP3T/PC$_{61}$BM-THF | 9.6 | 1 | $8.0 \times 10^{-6}$ | 0.294 | $5.6 \times 10^5$ |
| $^{#3}$ T-PDPP3T/PC$_{61}$BM | 8.2 | 25 | $1.1 \times 10^{-5}$ | 0.398 | 376 |
| $^{#4}$ T-PDPP3T/PC$_{61}$BM-SVA | 33 | 4 | $1.8 \times 10^{-6}$ | 0.336 | $8.3 \times 10^6$ |

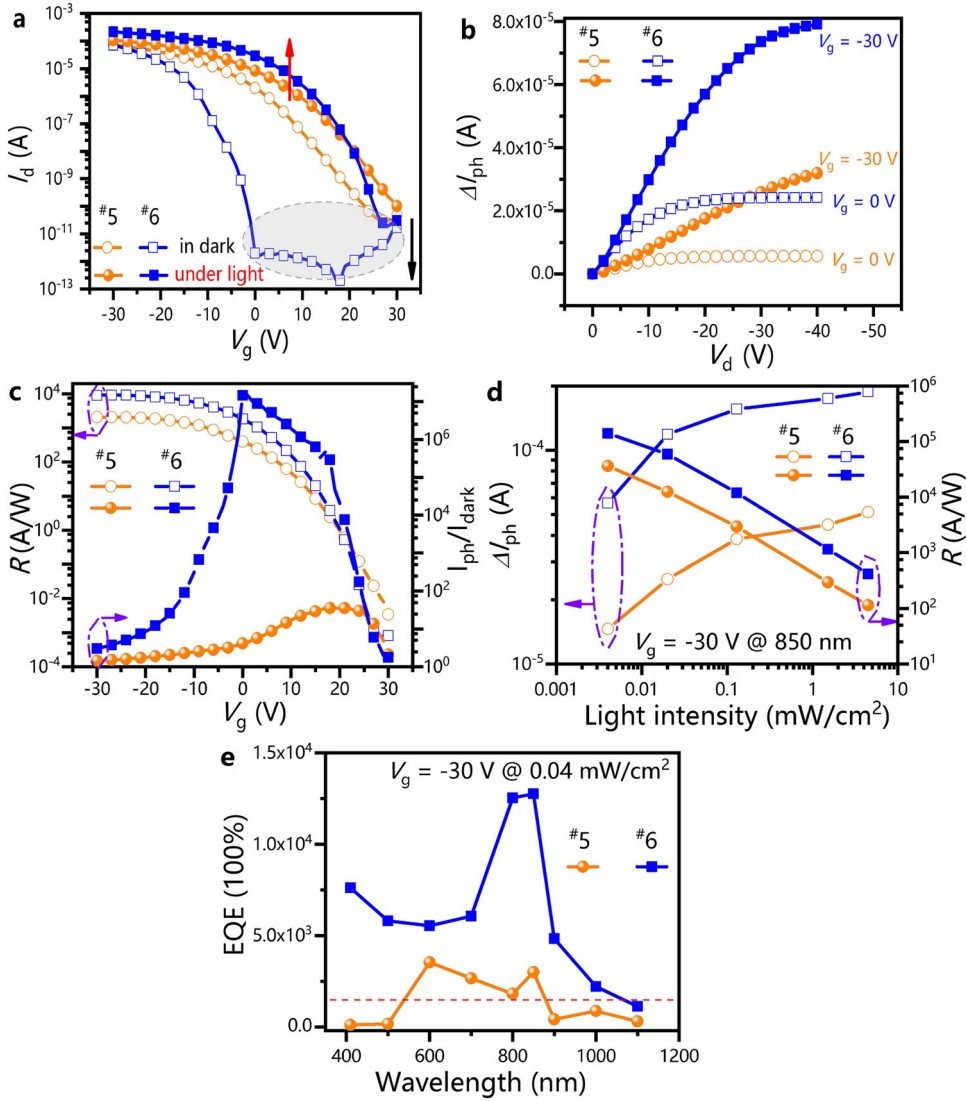

**Fig. 5 Electrical properties of devices with SVA (#6) and without SVA (#5) treatment. a** Transfer curve. **b** Photocurrent ($\Delta I_{ph}$) at various gate voltages. **c** Dependence of $R$ and $I_{ph}/I_{dark}$ on gate voltage. **d** Dependence of $\Delta I_{ph}$ and $R$ on light intensity (850 nm). **e** EQE spectrum at −30 V gate voltage under illumination of 0.04 mW/cm². The light intensity is shown in Fig. 5a–c, and Fig. 5e is 0.04 mW/cm² @ 850 nm. The orange lines with both unfilled and filled circles represent Device #5, and the blue lines with both unfilled and filled squares represent Device #6. The transfer curves of the devices were measured at a constant $V_d = -30$ V.

**Table 3 Performance parameters of the devices with and without SVA treatment.**

| | $\Delta I_{ph}$ (µA) | $V_o$ (V) in the dark | Off-state current (µA) | Hole mobility (cm²·V⁻¹·s⁻¹) | EQE | $R$ (A/W) | $I_{ph}/I_{dark}$ | $t_r$ (µs) | $t_f$ (µs) |
|---|---|---|---|---|---|---|---|---|---|
| #5 W/O SVA | 32 | 26 | $2.8 \times 10^{-5}$ | 0.464 | 3000 | 2000 | 22 | 261 | 84 |
| #6 With SVA | 79 | 0 | $1.7 \times 10^{-6}$ | 0.752 | 13000 | 8700 | $1.7 \times 10^7$ | 93 | 74 |

400–1100 nm (Fig. 5e, Supplementary Fig. 14b, Supplementary Fig. 15b), indicating that this device could also be used as a wide spectral response phototransistor.

The response speed of the devices before and after SVA was tested (Fig. 6a, b, Supplementary Figs. 16–17). The long lifetime of the phototransistor carriers resulted in a slow response speed since the turn-on voltages could hardly recover back to the initial value when changing the light on/off state[21,39]. The on/off ratios of photoswitching at various gate voltages were very small, and the light-off currents increased with increasing test time, leading to an increase in the fall time ($t_f$) to over 80 s (Supplementary

Fig. 17a, b). The above results indicated that for application as photoswitches, phototransistors had little advantage over photodiodes. A dual control measurement was applied to enhance the response speed of phototransistors[40]. With this method, the on/off switching ratio was greater than $1.0 \times 10^4$. The light-off currents were maintained at approximately $1.0 \times 10^{-9}$ A (Supplementary Fig. 17d). The rise/fall times of the SVA devices (#6) reached 93/74 µs (Fig. 6b). The response speed of the SVA devices (#6) was much faster than that of the device without SVA (#5, 261/84 µs, Fig. 6b), which could be attributed to the fact that the acceptor phases in the diffusion interface layer trapped

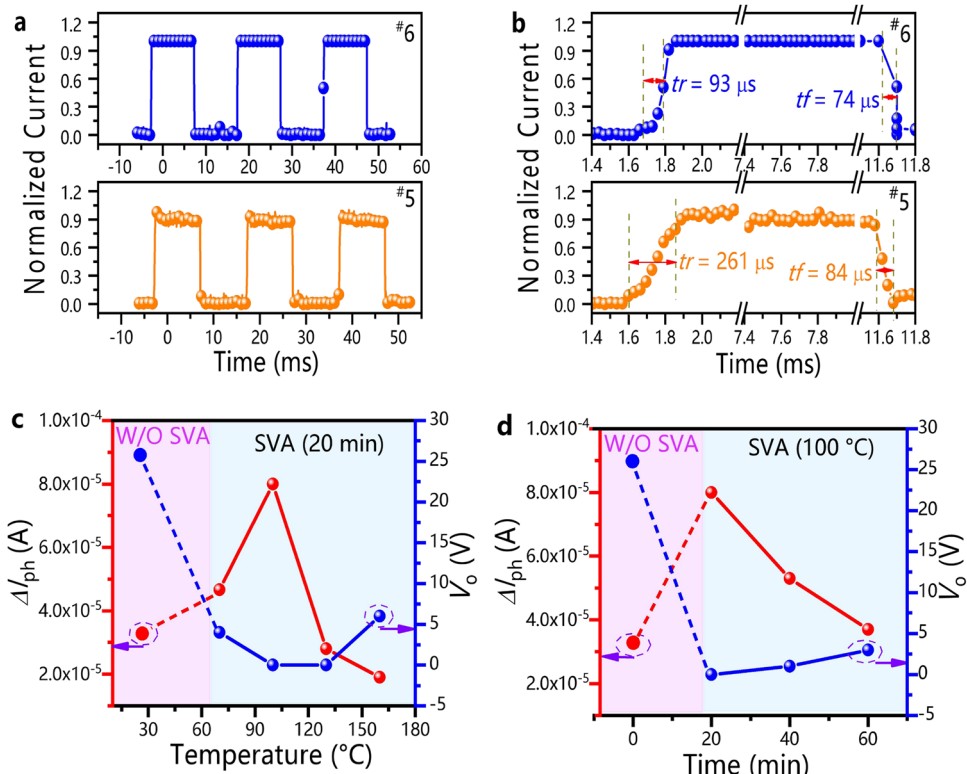

**Fig. 6 Time responses and variation in electrical properties under different SVA treatment conditions. a, b** Time responses via the photoelectric dual-control measurement (orange line filled with circles for Device #5, blue line with filled circles for Device #6, $tr$ for rise time, $tf$ for fall time). **c** Variation in $\Delta I_{ph}$ (red line with filled circles) and $V_o$ (blue line with filled circles) with increasing SVA temperature (light purple area represents the device without SVA, light blue area represents the devices with SVA (20 min)). **d** Variation in $\Delta I_{ph}$ (red line with filled circles) and $V_o$ (blue line with filled circles) with increasing SVA time (light purple area represents the device without SVA, light blue area represents the devices with SVA (100 °C)). The light intensity is 0.04 mW/cm² @ 850 nm.

photogenerated electrons and accelerated the response. It is worth noting that the photoelectric dual control method could also be applied as switches in ultrahigh-performance logical circuits. This method kept the gate voltage change and the light on/off state change at the same frequency, improving the response speed and stability of the switches. The use of this dual-control method also enabled organic materials with low mobility to be applied in logic circuits that required a large on/off switching ratio, fast response speed and high stability.

The acceptor phases in the diffusion interface layer could be controlled by changing the time interval and temperature of the SVA treatments (Fig. 6c, d, Supplementary Figs. 11–12, Supplementary Figs. 18–19). For example, with increasing SVA temperature, the turn-on voltages of the devices first decreased and then increased. However, the values were still significantly lower than the turn-on voltages of the devices without SVA (Fig. 6c). In addition, the trend of the device performance with extension of the SVA time is similar to that with the SVA temperature change (Fig. 6d). Combined with the information illustrated in Fig. 3 and Supplementary Figs. 4–5, it could be found that increasing the SVA temperature and extending the SVA time promoted the diffusion of more acceptor phases. Hence, diffusion interface layers with different thicknesses were formed, which ensured that the $V_o$ value of the devices with SVA treatment under different conditions was lower than that of the devices without SVA.

The performance of devices treated with SVA was better than those treated without SVA due to the existence of the diffusion interface layer. This phenomenon could be explained by the movement of the photogenerated carriers. For this purpose, the simplified physical model was used to illustrate the movement of

these carriers under two-electrode conditions (without the gate electrodes, Fig. 7). It is worth mentioning that the effect of the gate voltages on the photocurrents is discussed in the following session. Figure 7 shows the energy level diagram to analyse the effects of the diffusion interface layer on the increase in device performance. The energy levels of the PDPP3T and PC$_{61}$BM materials illustrated in this diagram were obtained by analyzing UPS and absorption spectra (Supplementary Table 1). For the SVA-free devices in the dark state (Fig. 7a), the electrons could be continuously injected from the gold electrodes into the channel transport layer, and a large number of electrons were generated[41]. During device operation, the recombination of electrons and holes would result in a large hysteresis window, high off-state dark currents, and turn-on voltages[42], which would adversely affect device performance (Supplementary Fig. 11b). However, for the devices with SVA in the dark state, the electrons injected by the electrode could transfer to the acceptor phases in the diffusion interface layer (Fig. 7c). Therefore, the channel transport layer would not be affected by the electrons, resulting in devices with a slight hysteresis window, small off-state dark currents and turn-on voltages, and high mobility (Supplementary Fig. 12b, Tables 2 and 3). According to Fig. 7b and d, when the devices were exposed to light, photogenerated electrons in the active layer would be captured by the acceptor phases, while the photogenerated holes would transfer to the channel transport layer for transmission. Moreover, the diffusion interface layer of the devices with SVA would also generate photogenerated carriers, resulting in additional photocurrents (Fig. 5b, d). As a result, the photogenerated currents, $I_{ph}/I_{dark}$ ratios, and response of the devices with SVA were greater than those of the devices without SVA (Table 3).

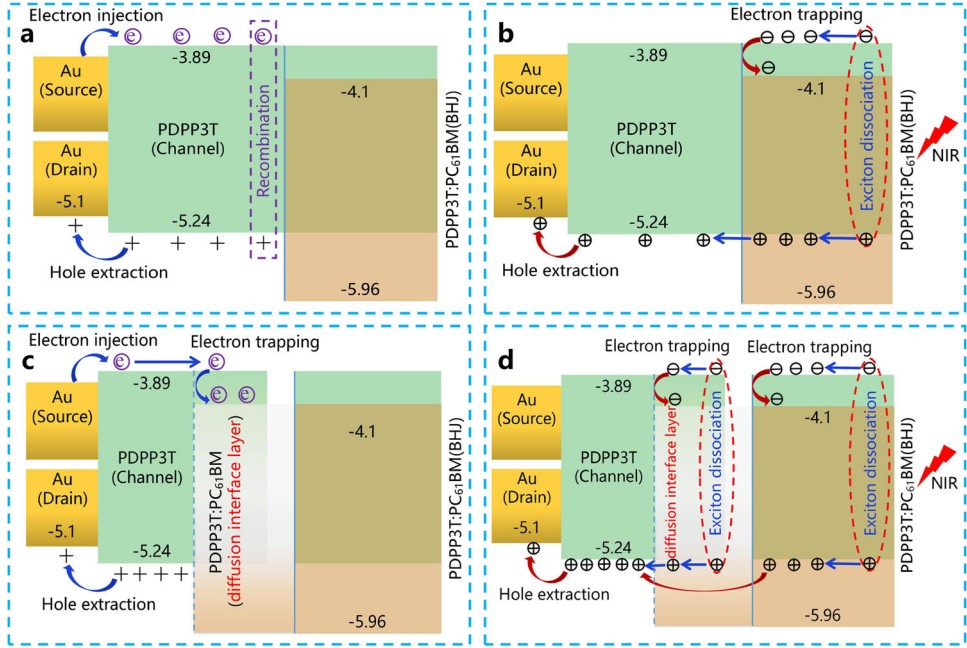

**Fig. 7 Energy level diagram of the different devices. a** Device without SVA in the dark state. **b** Device without SVA under illumination. **c** Device with SVA in the dark state. **d** Device with SVA under illumination.

Based on the analysis of the carrier movement under the condition of two electrodes (Fig. 7), the effects of gate voltages were discussed to explore the photogating/amplification effects on the performance of the phototransistors (Supplementary Figs. 20–21). When phototransistors are in the turn-off state, the photocurrents usually increase linearly with increasing light intensity due to the photoconductivity effect prevalent in photodiodes[43,44]. In comparison, when phototransistors are in the turn-on state, the photovoltaic effect is significant (Eq. 4) since the photovoltage is induced by the accumulation of trapped electrons[43]. It can be seen that the photocurrents varied in the devices with SVA (#6) and without SVA treatment (#5). The photocurrents exhibited a nonlinear relationship with the light intensity in the turn-on state ($V_g = -30$ V, Supplementary Figs. 20–21), which satisfied Eq. (4). Therefore, the transconductance ($G_M$) values of the devices were calculated to estimate the photogating/amplification effect. Due to the formation of a diffusion interface in devices (#6) after SVA treatment, the $G_M$ value of devices (#6) was 5.3 μS under weak light, which was lower than that in the devices without SVA (#5). The smaller $G_M$ value in Device #6 could be attributed to the occurrence of a large drift in the threshold voltages under weak light (Fig. 5a and Table 3). The $G_M$ values of the SVA devices (#6) increased slowly with increasing light intensity from 5.3 to 6.7 μS. In contrast, the $G_M$ value decreased from 7.3 to 4.3 μS in devices without SVA (#5). According to Fig. 7, the diffusion interface of the SVA devices (#6) provided an additional donor/acceptor interface to separate photogenerated carriers. As a result, under strong light, the photocurrents of the SVA devices (#6) were difficult to reach a saturated state (Supplementary Table 5), resulting in an increasing $G_M$ value (Supplementary Fig. 21b). The advantages of the diffusion interface layer were highlighted by comparing the changes in the $G_M$ value of the two devices.

## Discussion

In this research, a broad spectral response (400–1100 nm) of NIR polymer phototransistors with high EQEs (1000–13000 × 100%) was achieved. In addition, the prepared phototransistors exhibited an ultrahigh $I_{ph}/I_{dark}$ value of $1.7 \times 10^7$, a small turn-on voltage of 0 V, and a fast response speed of 93/74 μs. A diffusion interface layer, instead of a mutual dissolution interface layer, was formed between the channel transport layer and bulk heterojunction layer to prevent the random distribution of acceptor phases. The acceptor phases in the diffusion interface could trap the electrons injected by the electrode in the dark state and supply additional photocurrents when illuminated. Thus, the off-state dark currents decreased by one order of magnitude, and the photocurrents increased over twofold. This novel device structure was constructed by combining FTM with an SVA treatment. This strategy is expected to be widely used for composite systems that dissolve well in solvents with low boiling points. The novel device structure proposed in this work has promising applicability for fabricating high-performance and low-cost phototransistors.

## Methods

**Materials**. Octadecyltrichlorosilane (OTS, 98%), isopropanol (99%), tetrahydrofuran (THF, 99%) and o-dichlorobenzene (ODCB, 99%) were all purchased from J&K Scientific. CHCl₃ (99.8%) and PC₆₁BM were purchased from Sigma–Aldrich and 1-Material, respectively. Poly{2,20-[(2,5-bis(2-hexyldecyl)-3,6-dioxo-2,3,5,6-tetrahydropyrrolo[3,4-c]pyrrole-1,4-diyl)dithiophene]-5,50-diyl-alt-thiophen-2,5-diyl} (PDPP3T), indaceno[1,2-b:5,6-b']dithiophene-co-2,1,3-benzothiadiazole (IDT-BT), poly(3-hexylthiophene-2,5-diyl) (P3HT) and PEDOT:PSS were purchased from Solarmer Materials. None of the commercially available chemicals were further purified. In the first layer, high n-doped silicon was used as the back-gate dielectric, and thermally grown SiO₂ (300 nm) was selected as the inorganic gate dielectric.

**Device fabrication**. Both the Si/SiO₂ substrates and the OTS (30 μL) were placed in a petri dish covered by aluminum foil and heated on a hot plate (140 °C, two hours) in a nitrogen environment. The self-assembled OTS layer was prepared after cleaning the extra OTS off the substrates by spin-coating a CHCl₃ solution (100 μL, 6000 rpm, 60 s). The PDPP3T (0.2 wt%) and PDPP3T:PC₆₁BM (0.2 wt%) solutions used for spin-coating were dissolved in mixed solvents of ODCB and CHCl₃ (volume ratio = 4:1)[45]. All the solutions were stirred for over 12 hours at 100 °C. PC₆₁BM (1 wt%) was dissolved in CHCl₃ and THF, and the PC₆₁BM layer was spin-coated at a speed of 2000 rpm for 60 s. The PDPP3T layer (70 nm) was spin-coated in two steps (700 rpm for 10 s and 1200 rpm for 60 s), and the PDPP3T:PC₆₁BM layer (60 nm) was spin-coated at 1200 rpm for 60 s. Both the PDPP3T and PDPP3T:PC₆₁BM layers were vacuum annealed at 100 °C (10 min) to eliminate trapping and obtain a uniform phase orientation in the film. The PEDOT:PSS layer (20 nm) used to test the acceptor absorption peak was spin-

coated (2000 rpm, 60 s) in a solution diluted with 50% isopropanol. The source and drain electrodes of the device were prepared by thermally evaporating a Au film (50 nm). The devices with SVA treatment were placed in a petri dish (30 μL $CHCl_3$) covered by aluminum foil and then annealed in a nitrogen glove box.

**In situ Raman spectra**. Si/$SiO_2$/OTS was used as the substrate, and the active layer was selected (PDPP3T, $PC_{61}BM$, PDPP3T:$PC_{61}BM$, PDPP3T/$PC_{61}BM$ and PDPP3T/PDPP3T:$PC_{61}BM$) to prepare related samples. The Raman test data at the same depth were selected for in situ Raman characterization of the samples annealed at different temperatures. PDPP3T-25 °C, $PC_{61}BM$-25 °C and PDPP3T:$PC_{61}BM$-25 °C were tested at 25 °C, and the PDPP3T/$PC_{61}BM$ and PDPP3T/PDPP3T:$PC_{61}BM$ films were tested under annealing conditions of 25–130 °C and 25–100 °C, respectively. The variation in the characteristic Raman peak of $PC_{61}BM$ at 1460 $cm^{-1}$ was chosen to analyse the effect of annealing on $PC_{61}BM$.

**FTM**. In terms of the devices prepared by FTM, $CHCl_3$ (low boiling point) was used as the solvent for the PDPP3T (0.2 wt%), IDT-BT (0.25 wt%), $PC_{61}BM$ (1 wt%), P3HT (0.25%) and PDPP3T:$PC_{61}BM$ (0.2 wt%) solutions. All these solutions were stirred over 12 h at 90 °C. During the film transfer process, an organic solution within 30 μL was dropped into a petri dish filled with deionized water. The completely transferred film was vacuumed for more than 20 min to remove excess water vapor. The PDPP3T and PDPP3T:$PC_{61}BM$ films were annealed in a glove box at 100 °C in a nitrogen atmosphere for 10 min.

The whole FTM process was recorded as a video and added to this work in Supplementary Movie 1. Moreover, a series of high-quality films were successfully prepared via FTM (Supplementary Fig. 22), such as PDPP3T, IDT-BT, PDPP3T:$PC_{61}BM$, and P3HT films. Among all these films, the quality of the $PC_{61}BM$ film with a smaller molecular weight was slightly poor. In addition, devices of different sizes could be fabricated by FTM. Taking the IDT-BT film as an example, the transferred film still showed high quality even though the device size was increased to $8 \times 9$ $cm^2$ (Supplementary Fig. 23c). A large-area flexible film could also be obtained via FTM (Supplementary Fig. 23b). These results indicated that the preparation of high-quality films via FTM was scalable and promising.

**Photodetector characterization**. The measurements of the film thicknesses and absorption spectra (UV–vis) were performed with a Dektak 150 instrument (Veeco) and Shimadzu UV-3100 spectrophotometer, respectively. The films were observed by atomic force microscopy (AFM), transmission electron microscopy (TEM), and scanning electron microscopy (SEM) with a Dimension edge instrument (Bruker), JEOL-2100F instrument (JEOL), and Sigma 300 instrument (Zeiss), respectively. The in situ Raman spectra and film surface photographs were obtained with an inVia Reflex instrument (Renishaw) and P40 Pro instrument (HUAWEI), respectively. Ultraviolet photoelectron spectroscopy (UPS) measurements were carried out with a Nexsa (Thermo Fisher) system using a He(I) excitation energy of 21.22 eV. The grazing incidence X-ray diffraction (GIXRD) was investigated by a Smart Lab III instrument (Rigaku). The electrical performance of the phototransistors was recorded in air using a four-semiconductor parameter analyser (Keithley 2636B) with a Cascade probe station. During the photoelectric dual-control measurement[40], the response time at the microsecond level was measured by a digital storage oscilloscope (2012B, Tektronix) combined with a lock-in amplifier (LIA-MV-150, Femto). Light illumination at different wavelengths (410–2400 nm, 20 Hz) was provided by a continuous spectrum light source (Opolette 355 LD), where the light intensity was tested by a laser power meter (header Ophir NOVA II and probe PD300-UV).

**Displayed equations**. In the dark state, the device mobility (μ) is calculated according to the following equation[46]:

$$\mu = \frac{2L}{WC_i}\left(\frac{\partial \sqrt{I_d}}{\partial V_g}\right)^2 \qquad (1)$$

where $L$ (40 μm) is the channel length, $W$ (1000 μm) is the channel width, $I_d$ is the source-drain current, $V_g$ is the gate voltage. And $C_i$ (11.5 nF/$cm^2$) is the capacitance per unit area of the $SiO_2$ dielectric caused by the negligible capacitance of the self-assembled OTS monolayer.

In addition, the responsivity ($R$) and the external quantum efficiency (EQE, where the value is equal to ($G$) of a photodetector can be characterized by the following equation[2,47]:

$$R = \frac{\Delta I_{Ph}}{P_{inc}} \qquad (2)$$

$$EQE = R \times \frac{h\upsilon}{q} \times 100\% \qquad (3)$$

where $\Delta I_{ph}$ is the photocurrent, $P_{inc}$ is the incident light power, $h$ is the Planck constant, $\upsilon$ is the frequency of light, and $q$ is the absolute value of electron charge.

The phototransistors satisfy the photovoltaic effect in the turn-on state, which satisfies the following equation[43]:

$$\Delta I_{ph} = G_M \Delta V_{th} \qquad (4)$$

where $G_M$ is the transconductance and $\Delta V_{th}$ is the shift in the turn-on voltage.

## Data availability
Source data for Fig. 2c–f, Fig. 3, Fig. 4j–l, Figs. 5 and 6, Tables 1–3, Supplementary Figs. 3 and 4, Supplementary Figs. 6–9, Supplementary Figs. 11 and 12, Supplementary Figs. 14–21 and Supplementary Tables 1–5 are provided as a source data file. Source data are provided with this paper.

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

## Acknowledgements
T.H. acknowleges support form the Hunan Provincial Natural Science Foundation of China (2019JJ50565), Scientific Research Fund of Hunan Provincial Education Department (18A461), Scientific Research Fund of Chenzhou (zdyf201908), Fund of Xiangnan University (2019XJ29), the Scientific Research Start-up Fund for High-level Talents in Xiangnan University, 2020 National Innovation and Entrepreneurship Training Program for College Students (S202010545034). The work of X.F.Z. is supported by the Natural Science Foundation of China (51801034, 52172067), and the Guangdong Province Outstanding Youth Foundation (2021B1515020038). The work of L.L. is supported by the Natural Science Foundation of China (51873068, 51573055).

## Author contributions
T.H. and X.Z. conceived the idea and supervised the project. Z.W., N.S. and Z.Z. fabricated the device and carried out optoelectronic characterizations. X.H. and L.L. contributed to the in-situ Raman tests and analysis. S.D. contributed to the response speed characterizations. C.J. and X.H. proposed the mescanism of the device. T.H. and X.Z. interpreted the results and wrote the manuscript.

## Competing interests
The authors declare no competing interests.
