## [Peer Review File · Nature Communications]

Peer review comments, first round review–

Reviewers' comments:

Reviewer #1 (Remarks to the Author):

Dear Mr. Han, dear Mr. Zhang,

congratulations to this interesting finding concerning the methodology in Polymer Phototransistor manufacturing.

The improvement in photosensitivity and especially the reduction in turn-on voltage of your devices are impressive.

At the same time I have to express my concern with certain aspects of the manuscript. There are two reasons that lead to the conclusion that I cannot recommend the publication of your manuscript in Nature Communications at this point.

1) While the method you found is very impressive, there is not enough scientific depth for a publication in such a high-ranking journal. This becomes apparent in your text, when assumptions about physical processes are made, but no experimental proof is given. Examples are:
* Line 125: "The TEM image shows that the phase separation in the transferred PDPP3T:PC 61 BM film is better than that in the spin-coated film" -> Such a statement needs to be explained. What does "better" mean in this context? How do you draw this conclusion from the images?
* Figure 4 a-c: How do you know the layer structure that you schematically depict in these figures? A cross-sectional SEM image could be very helpful here, for example. I believe the experiment would show significantly different structures especially around the electrodes. Interfacial solvent diffusion may attack the electrode-semiconductor interface and explain the large turn-on voltages you observe.

2) I do not find the overall quality of the manuscript sufficient for the journal. This concerns several aspects:

- * Proof-reading: Please have a native speaker of the English language correct your manuscript. Especially indefinite and definite articles as well as plural and singular are misused in almost every sentence. This made it quite difficult for me to understand your interesting work.
- * The logical structure of the text should be improved to guide the reader through your text, rather than listing your findings one by one.
- * Many of the figures are difficult to understand. Here are some exemplary suggestions for Figure 2:
 - ** Fig 2b: Please add a scalebar. Please denote the lower end of the color-legend.
 - ** Fig 2c/d: Please improve the legibility of the graphs by using more consistent fonts and font-sizes, avoiding very light colors and clearly separating the two graphs from each other. Also, please start the y-axis in Fig 2c at 0, since negative absorption values are not reasonable.

With these improvements, I think that your manuscript will be a very valuable contribution to the community. I hope you can find a suitable journal for this nice work and wish you all the best for the publication process.

Best regards,
Felix Dollinger

--

Dr. Felix Dollinger
IAPP - Dresden Integrated Center for Applied Physics and Photonic Materials
Technische Universität Dresden

Reviewer #2 (Remarks to the Author):

In this report, Han et. al. report a bilayer organic phototransistor designed to detect near infrared light (NIR). NIR photodetectors based on organic semiconductors are a well-motivated and interesting technology, and understanding the physics of this bilayer system could be of interest to the community. However, I am unable to recommend this manuscript for publication in Nature Communications. The authors have focused on reporting a high-performance device but did not provide a convincing argument why their device is likely to be a superior light sensing strategy than the more commonly employed photodiodes. In particular, the device fabrication process is complex and decidedly not scalable, and I was not convinced of the merits of these devices from a technological point of view. I could potentially see the value in the report from the point of view of understanding the physics of bilayer organic phototransistors, but there was not a sufficiently convincing argument for any of the conclusions made. The manuscript generally focused on reporting high quantitative metrics, however I don't think this is sufficient alone to warrant publication in Nature Communications.

One metric of concern is responsivity. Responsivity in phototransistors should be interpreted very carefully. It is a parameter originally used to quantify the performance of photodiodes: devices without amplification. It has here been directly applied to phototransistors and I'll explain why this is a problem. In transistors, the denominator in the equation for R contains the product of the device length * width (LW) multiplied by optical power density. However, unlike a photodiode, for a transistor L/W determines the magnitude of source drain current (I_D). As an example, consider two otherwise identical transistors: one which is square ($L=W$) and one with a very short channel length and long channel width $W \gg L$, but where the product (LW) was the same for both. The former would have a lower I_D than the latter, but because they have the same LW , the denominator in R is the same. Now, if the illumination were to shift the threshold voltage by roughly the same amount in each, one would expect a much greater change in I_D for the narrow device than the square device. I.e. the shape of the transistor will almost certainly affect R . This is not the case for photodiodes because current flows perpendicular to the plane of LW and, broadly speaking, the shape of the device area doesn't matter too much as long as LW is the same. I am not trying to single out the authors here. I aware that pretty much every phototransistor paper contains R evaluated in this way. What I am saying is that reporting a phototransistor with an enormous R is not a noteworthy achievement on its own: you can achieve this by making the transistor long and thin.

It's also not clear to me why the gate and light co-control method depicted in Figure S13 was used? Surely it would make sense to test the device response to light only? If the device were to be used as a phototransistor you would not know when light was about to be incident on the device. Measuring the response to light alone is a very powerful tool in phototransistors as it can give a hint as to what extent current enhancement is coming from threshold voltage induced shifts vs directly generated charges. The former process is generally slower than the latter. To me the only benefit in doing the measurement this way is the ability to report a parameter with a high speed, which itself is not sufficient to warrant publication in a high impact journal like this. Also, it's also not clear why the current levels were normalized.

There is insufficient direct evidence for the hypothesized mechanisms. I am more or less willing to accept the schematics in 4(a-c) are reasonable. However, I cannot accept that the process in Scheme 1 can be concluded, with this level of detail, from device measurements, Raman, UVVis, and AFM/SEM alone. The authors state "The effects of the diffusion interface layer on the device performance are analyzed with an energy level diagram as shown in Scheme 1." But as far as I can tell, this diagram was not measured, and is more or less entirely hypothesized. In particular, one of the biggest challenges with phototransistors is trying to disentangle the amplification (essentially photo-gating) vs charges being directly photogenerated and contributing to current (like in a solar cell). This is important because changes in threshold voltage are generally slow, but give rise to large EQE (often >100%). It is clear that photogating / amplification is playing a role from the big shifts in threshold voltage (S10 (b) is a good example) but no distinction was made between this process and the current generation process hypothesized in Scheme 1.

I also have a few more minor comments:

In the introduction these two statements need to be elaborated upon or provided with a reference:

- "However, the acceptor phase in traditional single layer bulk heterojunction (BHJ) photodetector

needs a high concentration, which easily causes electron/hole recombination.”

- “It is reported that bilayer devices (channel transport layer/ BHJ layer) can effectively reduce the effects of acceptors on device performance”

Figure 2(b) needs a lateral scale bar. The peaks in Figure 2(d) need to be labeled in the caption. The caption to Figure 2(d) also should explain what is being shown since this is not what one expects when seeing a “GIWAXS image”. I assume this is a line cut. The text mentions a different abbreviation “GIXRD” to “GIWAXS” in the caption. In either case, the abbreviation needs to be spelled out on first use.

The labels “0-0”, “0-1” and “1-0” in 3(c) need to be explained in the caption.

The section referring to Raman needs to be re-written to be clearer. It was very difficult to follow in its current form.

Why was turn on voltage (V_{0}) used rather than the more conventional threshold voltage (V_{Th})?

I don’t agree with the conclusion that “The hysteresis window of the FTM-based bilayer device 6# obtained after SVA reduced significantly, indicating that the defects/traps in the film could be effectively eliminated...” hysteresis is not the only possible manifestation of trap states.

Be aware that intrinsic has a very specific meaning in semiconductors and using the phrase “intrinsic electrons” so frequently may be confusing to the general reader.

Manuscript ID: *NCOMMS-21-31071*

Title: **Diffusion Interface Layer Controlling the Acceptor Phase for Ultrahigh Photosensitivity Bilayer Near-Infrared Polymer Phototransistors**

Journals: **Nature Communications**

Reviewers' comments:

Reviewer #1 (Remarks to the Author):

Congratulations to this interesting finding concerning the methodology in Polymer Phototransistor manufacturing. The improvement in photosensitivity and especially the reduction in turn-on voltage of your devices are impressive. At the same time I have to express my concern with certain aspects of the manuscript. There are two reasons that lead to the conclusion that I cannot recommend the publication of your manuscript in Nature Communications at this point.

Author replay:

Thanks for the reviewer's suggestion. The revisions are highlighted by red color in the revised manuscript.

1. While the method you found is very impressive, there is not enough scientific depth for a publication in such a high-ranking journal. This becomes apparent in your text, when assumptions about physical processes are made, but no experimental proof is given. Examples are:

(1) Line 125: "The TEM image shows that the phase separation in the transferred PDPP3T:PC₆₁BM film is better than that in the spin-coated film" -> Such a statement

needs to be explained. What does "better" mean in this context? How do you draw this conclusion from the images?

Author replay:

Thanks for the reviewer's suggestion. There are a wrong expression here, which should be changed into "The TEM image shows that the phase separation in the spin-coated PDPP3T:PC₆₁BM film is better than that in the transferred film". The reason is given in the below:

The TEM was used to observe the phase separation in the organic blend films [Choi, Y. et al. *Nano Energy* **30**, 200-207 (2016)]. In the TEM images, PC₆₁BM appeared dark due to its high electron density, while PDPP3T appeared bright (Figure 2a). The TEM images of transferred PDPP3T:PC₆₁BM films had more dark areas than that of the spin-coated films (Figure S2). This phenomenon suggests that the separation of the donor and acceptor phases in transferred films was poorer, the reason of which can be that more aggregated PDPP3T phases were formed in the transferred films, thus imposing less effect on the PC₆₁BM phases. Additionally, the poor phase separation in transferred PDPP3T:PC₆₁BM films was also confirmed by the observation under atomic force microscopy (AFM) in the (Figure 2b). We supplemented it in the revised manuscript (see in page 6, last paragraph).

Albeit having low acceptor phases concentration, the blend films with poor phase separation can already produce an obvious light response in the phototransistors due to the existence of gate electrodes. Therefore, the high crystallization of PDPP3T phases formed in the transferred films would ensure big hole mobility, further bringing large photocurrents of the devices. We supplemented it in the revised manuscript (see in page 7, first paragraph).

Figure S2. Locally enlarged TEM images of the active layer in the Figure 2a: (a) spin-coated PDPP3T:PC₆₁BM, (b) transferred PDPP3T:PC₆₁BM.

(2) Figure 4 a-c: How do you know the layer structure that you schematically depict in these figures? A cross-sectional SEM image could be very helpful here, for example. I believe the experiment would show significantly different structures especially around the electrodes. Interfacial solvent diffusion may attack the electrode-semiconductor interface and explain the large turn-on voltages you observe.

Author replay:

Thanks for the reviewer's suggestion. We have supplemented the cross-sectional SEM imaging of different devices without Au electrodes (Figure 4g-i). This is because the large electrode spacing ($\sim 40 \mu\text{m}$) hinders the observation of the details of each layer when the complete conductive channels are all present in the field of view. The features of these films were described by the schematic diagram shown in Figure 4a-c. For the spin-coated bilayer devices, when using the CHCl_3 -soluble PC₆₁BM solution, the interface between PDPP3T and PC₆₁BM films was almost invisible ($\sim 184 \text{ nm}$, Figure 4g), indicating a high degree of mutual dissolution of these two films. In comparison, when the THF-soluble PC₆₁BM solution was used, the relatively blurred interface between the PDPP3T and the PC₆₁BM can be observed (Figure 4h),

indicating the partial dissolution of these two films. Besides, an obvious interface was observed in the transferred bilayer devices (Figure 4i), suggesting no mutual dissolution between the first and the second layer. In words, combined the cross-sectional morphology (Figure 4g-i) with the film surface photographs (Figure 4d-f), it can be verified that the schematic diagram of bilayer films prepared with different methods is reasonable (Figure 4a-c). We supplemented it in the revised manuscript (see in page 12, first paragraph).

In addition, the Au, as the electrodes, has stable chemical properties. During the spin-coating process of the second layer (PC₆₁BM film), the solution evaporates quickly, ensuring the short contact time (< 60 s) between the Au electrodes and chloroform (or tetrahydrofuran). Instead, the chloroform solvent has completely volatilized when transferring the PC₆₁BM film. Hence, it is believed that the Au electrodes are less likely to be attacked by the solvent in the above methods.

Figure 4. Schematic diagram of devices with Si/SiO₂/ OTS/ PDPP3T/ Au/ PC₆₁BM

structure via different fabrication methods: (a) PC₆₁BM solution dissolved in chloroform solvent for spin-coating bilayer device (1[#] S-PDPP3T/ PC₆₁BM-CHCl₃), (b) PC₆₁BM solution dissolved in THF solvent for spin-coating bilayer device (2[#] S-PDPP3T/ PC₆₁BM-THF), (c) bilayer device via FTM (3[#] T-PDPP3T/ PC₆₁BM). The film surface photographs: (d) 1[#] S-PDPP3T/ PC₆₁BM-CHCl₃, (e) 2[#] S-PDPP3T/ PC₆₁BM-THF, (f) 3[#] T-PDPP3T/ PC₆₁BM. The cross-sectional SEM imaging of the films: (g) 1[#] S-PDPP3T/ PC₆₁BM-CHCl₃, (h) 2[#] S-PDPP3T/ PC₆₁BM-THF, (i) 3[#] T-PDPP3T/ PC₆₁BM.

2. I do not find the overall quality of the manuscript sufficient for the journal. This concerns several aspects:

(1) Proof-reading: Please have a native speaker of the English language correct your manuscript. Especially indefinite and definite articles as well as plural and singular are misused in almost every sentence. This made it quite difficult for me to understand your interesting work.

Author replay:

Thanks for the reviewer's suggestion. We have studied the reviewer's literatures well, and learned how to use the indefinite and definite articles properly [Dollinger, F. et al. *Adv. Mater.* **31**, 1900917 (2019); Dollinger, F. et al. *Adv. Electron. Mater.* **5**, 1900576 (2019)]. Indeed, reviewer's writing method was very rigorous and benefited our manuscript revision a lot. The misleading expressions have been changed into clearer ones, such as:

(a) the original sentence "As a result, the photodetector usually has low mobility, high off-state dark current and large turn-on voltage (V_o)" changes into "As a result, the phototransistors usually have low mobility, high off-state **dark currents**, and large **turn-on voltages** (V_o)" (see in page 3, the first paragraph).

(b) the original sentence "It is reported that bilayer devices (channel transport layer/

BHJ layer) can effectively reduce the effects of acceptors on device performance” changes into “It **has been** reported that **the design of** bilayer devices (channel transport layer/ BHJ layer) **could** effectively **enhance** devices performance” (see in page 3, the last paragraph).

- (c) the original sentence “The dropped solution expanded rapidly on the deionized water surface” changes into “The dropped solution expanded rapidly on deionized water surface **to form a film**” (see in page 5, the last paragraph).

(2) The logical structure of the text should be improved to guide the reader through your text, rather than listing your findings one by one.

Author replay:

Thanks for the reviewer's suggestion. We have modified the logical structure between sentences, and we have added a lot of analysis as well, such as:

- (a) Application prospect of photoelectric dual control method have added: “It is worth noticing that the photoelectric dual control method can also be applied as switches in ultrahigh-performance logical circuit. The response speed and stability of the switches can be improved by this method keeping the gate voltage change and the light on/off state change at the same frequency. The employment of this dual control method also enables the use of organic materials with low mobility in logic circuits that require large switching on/off ratio, fast response speed, and high stability”. We supplemented it in the revised manuscript (see in page 18, first paragraph).
- (b) The discuss of the phase separation: “Albeit having low acceptor phases concentration, the blend films with poor phase separation can already produce an obvious light response in the phototransistors due to the existence of gate electrodes. Therefore, the high crystallization of PDPP3T phases formed in the transferred films would ensure big hole mobility, further bringing large

photocurrents of the devices”. We supplemented it in the revised manuscript (see in page 7, first paragraph).

(c) The discuss of the responsivity (R): “In general, the R value of the phototransistors tended to be large (~ 8700 A/W, Table 3) due to the photogating effect. The device with a high W/L value usually has a high R value. Compared with W/L value of the device with ultrahigh R value [Guo, Y. et al. *Adv. Funct. Mater.* **20**, 1019-1024 (2010); Chow, P. C. Y. et al. *Nat. Commun.* **9**, 1-8 (2018)], the W/L value in our device was lower. Therefore, the R value increased in the phototransistors prepared in this work might not relate to the high W/L value, but could be caused by the rise in the number of the photo-generated carriers”. We supplemented it in the revised manuscript (see in page 16, first paragraph).

(3) Many of the figures are difficult to understand. Here are some exemplary suggestions for Figure 2:

1) Fig 2b: Please add a scalebar. Please denote the lower end of the color-legend.

Author replay:

Thanks for the reviewer’s suggestion. We supplemented the scalebar of AFM diagram in Figure 2b, and its scanning range is $2 \times 2 \mu\text{m}^2$ (see in page 9, Figure notes in Figure 2).

Figure 2. The TEM (a), AFM (b) of different films.

2) Fig 2c/d: Please improve the legibility of the graphs by using more consistent fonts and font-sizes, avoiding very light colors and clearly separating the two graphs from each other. Also, please start the y-axis in Fig 2c at 0, since negative absorption values are not reasonable.

Author replay:

Thanks for the reviewer's suggestion. We changed the light colored lines into the dark ones in Figure 2c-d, and the y-axis in Figure 2c was modified to display from 0.

Figure 2. The UV-vis absorption spectra (c) and grazing-incidence X-ray diffraction (GIXRD) patterns (d) of different films.

Reviewer #2 (Remarks to the Author):

1. In this report, authors et. al. report a bilayer organic phototransistor designed to detect near infrared light (NIR). NIR photodetectors based on organic semiconductors are a well-motivated and interesting technology, and understanding the physics of this bilayer system could be of interest to the community. However, I am unable to recommend this manuscript for publication in Nature Communications. The authors have focused on reporting a high-performance device but did not provide a convincing argument why their device is likely to be a superior light sensing strategy than the more commonly employed photodiodes.

Author replay:

Thanks for the reviewer's suggestion. We have supplemented a comparison between photodiodes and phototransistors in the introduction. The phototransistors are able to detect much weaker light signals than the photodiodes because the light can directly reach the active layer of the phototransistors. At the same voltages, the dark currents of the photodiodes are larger than that of the phototransistor since the electrode distance (< 500 nm) of the photodiodes is much shorter (phototransistors: > 5 μm) [Wang, H. et al. *Chem. Soc. Rev.* **46**, 5204-5236 (2017); Guo, F. et al. *Nat. Nanotechnol.* **7**, 798-802 (2012)]. It is more challenging to reduce the dark currents of photodiodes than that of phototransistors. The dark currents of photodiodes can be reduced by hindering the injection of carriers, which is usually realized by either forming additional electron/hole blocking layers or optimizing the energy alignment at each interface. In contrast, the phototransistors can utilize additional gate electrodes to effectively suppress the dark currents and amplify the photocurrents. We supplemented it in the revised manuscript (see in page 2, last paragraph).

In addition, we have added a comparison between photodiodes and phototransistors in the Results and Discussion, for example:

- (a) “The above results indicate that for the application as photoswitches, phototransistors almost have no advantage over photodiodes (see in page 17, last paragraph)”.
- (b) “When the phototransistors are in the turn-off state, the photocurrents usually increase linearly with the increase of light intensity due to a photoconductivity effect, which commonly exists in the photodiodes [Noh, Y.-Y. et al. *J. Appl. Phys.* **100**, 094501 (2006); Fuentes-Hernandez, C. et al. *Science* **370**, 698-701 (2020)].” We supplemented it in the revised manuscript (see in page 21, second paragraph).

2. In particular, the device fabrication process is complex and decidedly not scalable, and I was not convinced of the merits of these devices from a technological point of view. I could potentially see the value in the report from the point of view of understanding the physics of bilayer organic phototransistors, but there was not a sufficiently convincing argument for any of the conclusions made. The manuscript generally focused on reporting high quantitative metrics, however I don't think this is sufficient alone to warrant publication in Nature Communications.

Author replay:

Thanks for the reviewer's suggestion. The whole FTM process was recorded, and the video was provided and added to this work as **supporting information 2**. Meanwhile, a series of high-quality films were prepared successfully via FTM (Figure S22), such as PDPP3T, IDT-BT, PDPP3T:PC₆₁BM, P3HT films, and so on. Among all these films, the PC₆₁BM film with a smaller molecular weight shows slightly poor film quality. In addition, devices of different sizes can be fabricated by FTM. Taking IDT-BT film as an example, the transferred film still shows high quality even though

the device size was increased to $8 \times 9 \text{ cm}^2$ (Figure S23c). A large-area flexible film can also be obtained via FTM (Figure S23b). These results indicate that using FTM to prepare high-quality films is scalable and promising. We supplemented it in the revised manuscript (see in page 25, last paragraph).

In addition, we have supplemented the cross-sectional SEM imaging of the films (Figure 4g-i) and ultraviolet photoelectron spectrometer (UPS) spectra to support physical mechanism in the Scheme 1. Meanwhile, the transconductance (G_M) of the phototransistors was calculated to explain the photogating/amplification effect (Figure S20-21).

Figure S22. (a) The different solutions expansion on the deionized water surface, (b) the different films surface photographs via FTM.

Figure S23. (a) The IDT-BT solution expansion on the deionized water surface, (b) the IDT-BT film photographs via FTM on the flexible substrate, (c) the IDT-BT film photographs with different sizes via FTM.

Figure 4. The SEM of the film cross sections: (g) 1[#] S-PDPP3T/ PC₆₁BM-CHCl₃, (h) 2[#] S-PDPP3T/ PC₆₁BM-THF, (i) 3[#] T-PDPP3T/ PC₆₁BM.

Figure 2. Ultraviolet photoelectron spectrometer (UPS) spectra of valence band region (e) and secondary electron cut-off region (f).

Figure S20. (a) Dependence of photocurrent (I_{ph}) and threshold voltage drift (ΔV_{th}) on light intensity of the 5[#]-W/O SVA at $V_g = -30$ V, (b) dependence of G_M on light intensity of the 5[#]-W/O SVA at $V_g = -30$ V.

Figure S21. (a) Dependence of photocurrent (I_{ph}) and threshold voltage drift (ΔV_{th}) on light intensity of the 6[#]-W/O SVA at $V_g = -30$ V, (b) dependence of G_M on light intensity of the 6[#]-W/O SVA at $V_g = -30$ V.

3. One metric of concern is responsivity. Responsivity in phototransistors should be interpreted very carefully. It is a parameter originally used to quantify the performance of photodiodes: devices without amplification. It has here been directly applied to phototransistors and I'll explain why this is a problem. In transistors, the denominator in the equation for R contains the product of the device length * width (LW) multiplied by optical power density. However, unlike a photodiode, for a transistor LW determines the magnitude of source drain current (I_D). As an example, consider two otherwise identical transistors: one which is square ($L=W$) and one with a very short channel length and long channel width $W \gg L$, but where the product (LW) was the same for both. The former would have a lower I_D than the latter, but because they have the same LW , the denominator in R is the same. Now, if the illumination were to shift the threshold voltage by roughly the same amount in each, one would expect a much greater change in I_D for the narrow device than the square device. I.e. the shape of the transistor will almost certainly affect R . This is not the case for photodiodes because current flows perpendicular to the plane of LW and, broadly speaking, the shape of the device area doesn't matter too much as long as LW is the same. I am not trying to single out the authors here. I aware that pretty much

every phototransistor paper contains R evaluated in this way. What I am saying is that reporting a phototransistor with an enormous R is not a noteworthy achievement on its own: you can achieve this by making the transistor long and thin.

Author replay:

Thanks for the reviewer's suggestion. I agree with the reviewer's understanding of the responsivity in phototransistors. However, the parameter we emphasized in this work is not responsivity. In general, the R value of the phototransistors tended to be large (~ 8700 A/W, Table 3) due to the photogating effect. The device with a high W/L value usually has a high R value. Compared with W/L value of the device with ultrahigh R value [Guo, Y. et al. *Adv. Funct. Mater.* **20**, 1019-1024 (2010); Chow, P. C. Y. et al. *Nat. Commun.* **9**, 1-8 (2018)], the W/L value in our device was lower. Therefore, the R value increased in the phototransistors prepared in this work might not relate to the high W/L value, but could be caused by the rise in the number of the photo-generated carriers. We supplemented it in the revised manuscript (see in page 16, first paragraph).

In the dark state, the turn-on voltages of the SVA devices ($6^\#$) could be reduced to almost zero by controlling the diffusion of the acceptor phases, suggesting that this device could be used at low operating voltages. In comparison, under illumination conditions (0.04 mW/cm², 850 nm), the turn-on voltages of the SVA devices ($6^\#$) increased to 27 V as a large response window was obtained (Figure S13). The $I_{\text{ph}}/I_{\text{dark}}$ value ($\sim 1.7 \times 10^7$ @ $V_g = 0$ V, Table 3) of the SVA devices ($6^\#$) was significantly higher than that of the device without SVA ($5^\#$). This $I_{\text{ph}}/I_{\text{dark}}$ value is higher than that of the NIR phototransistors reported in the literature [Li, F. et al. *Adv. Electron. Mater.* **3**, 1600430 (2016); Li, D. et al. *Adv. Funct. Mater.* 2105887, (2021)]. The high $I_{\text{ph}}/I_{\text{dark}}$ value of our phototransistors can be attributed to the negative shift of the turn-on voltages. On the one hand, the dark currents maintain at a low level with the aid of the gate electrodes; on the other hand, the photocurrents increase with the negative shift of the turn-on voltages. We supplemented it in the revised manuscript (see in page 16, last paragraph).

Figure S13. The effect of threshold voltage drift on the value of I_{ph}/I_{dark} and response window.

4. It's also not clear to me why the gate and light co-control method depicted in Figure S13 was used? Surely it would make sense to test the device response to light only? If the device were to be used as a phototransistor you would not know when light was about to be incident on the device. Measuring the response to light alone is a very powerful tool in phototransistors as it can give a hint as to what extent current enhancement is coming from threshold voltage induced shifts vs directly generated charges. The former process is generally slower than the latter. To me the only benefit in doing the measurement this way is the ability to report a parameter with a high speed, which itself is not sufficient to warrant publication in a high impact journal like this. Also, it's also not clear why the current levels were normalized.

Author replay:

Thanks for the reviewer's suggestion. The long lifetime of the phototransistors carriers results in slow response speed since the turn-on voltages can hardly recover back to the initial value when changing the light on/off state [Jin, Z. et al. *ACS Appl. Mater. Inter.* **8**, 33043-33050 (2016)]. The on/off ratios of photoswitching at various gate voltages were very small, and the light-off currents increased with the prolonging

of test time, leading to an increase in the fall time (tf) to over 80 s (Figure S17a-b). The above results indicate that for the application as photoswitches, phototransistors almost have no advantage over photodiodes. In addition, the photoswitching performances without gate voltage (two-terminal device) and 0 V gate voltage (three-terminal device) were compared, which showed the existence of gate voltage could suppress light-off current (Figure S17c). A dual control measurement was applied to enhance the response speed of phototransistors. With this method, the switching on/off ratio was over 1×10^4 . The light-off currents were maintained at around 1×10^{-9} A (Figure S17d). We supplemented it in the revised manuscript (see in page 17, last paragraph).

It is worth noticing that the photoelectric dual control method can also be applied as switches in ultrahigh-performance logical circuit. The response speed and stability of the switches can be improved by this method keeping the gate voltage change and the light on/off state change at the same frequency. The employment of this dual control method also enables the use of organic materials with low mobility in logic circuits that require large switching on/off ratio, fast response speed, and high stability. We supplemented it in the revised manuscript (see in page 18, first paragraph).

Besides, the reviewer pointed out that measuring the response to light alone is a very powerful tool to judge the source of the photocurrent enhancement. In fact, this indicator can be easily obtained from the I_{ph}/I_{dark} value on the transfer curve. The phototransistors satisfy the photovoltaic effect in the turn-on state, which meet the following equation [Noh, Y.-Y. et al. *J. Appl. Phys.* **100**, 094501 (2006)]:

$$I_{ph} = G_M \Delta V_{th} \quad (4)$$

It can be seen that the photocurrents varied in the devices with SVA (6[#]) and without SVA treatment (5[#]). The photocurrents exhibited a non-linear relationship with the light intensity in turn-on state ($V_g = -30$ V, Figure S20-21), which satisfied equation (4). Therefore, the transconductance (G_M) values of the devices were

calculated to estimate the photogating/amplification effect. The G_M values of the SVA devices ($\sigma^\#$) increased slowly from 5.3 to 6.7 μS with the increase of light intensity (see in page 21, last paragraph). Since the light-off currents (I_{dark}) of the device can be suppressed to a low level by the gate voltage, the $I_{\text{ph}}/I_{\text{dark}}$ value increases with the ΔV_{th} increasing (equation 4). In other words, the current enhancement from threshold voltage can be quantitatively described, indicating the parameter of photoswitch is not necessary.

At last, during the photoelectric dual control measurement, the response time at the microsecond level was measured by a digital storage oscilloscope (2012B, Tektronix), combined with a lock-in-amplifier (LIA-MV-150, Femto) (see in page 26, first paragraph). The measure principle of these instruments is to judge the device on/off state by high/low electric potential, which shows the normalized current levels.

Figure S17. The time-current curves of $\sigma^\#$ -with SVA device at a constant source drain voltage ($V_d = -30$ V) under light excitation ($0.04 \text{ mW/cm}^2 @ 850 \text{ nm}$): (a) multiple switching cycles with different gate voltage (V_g); (b) single switching cycle with

different gate voltage (V_g); (c) with $V_g = 0$ V (three-terminal device) and without gate voltage (two-terminal device); (d) the gate and light co-control method testing.

Figure S20. (a) Dependence of photocurrent (I_{ph}) and threshold voltage drift (ΔV_{th}) on light intensity of the 5[#]-W/O SVA at $V_g = -30$ V, (b) dependence of G_M on light intensity of the 5[#]-W/O SVA at $V_g = -30$ V.

Figure S21. (a) Dependence of photocurrent (I_{ph}) and threshold voltage drift (ΔV_{th}) on light intensity of the 6[#]-W/O SVA at $V_g = -30$ V, (b) dependence of G_M on light intensity of the 6[#]-W/O SVA at $V_g = -30$ V.

Table S5. Performance parameters of devices with and without SVA treatment.

Light intensity (mW/cm ²)	0.004	0.02	0.13	1.52	4.5
ΔV_{th} (V) - 5 [#] W/O SVA	2	5	8	10	12
ΔV_{th} (V) - 6 [#] With SVA	10.5	19.5	24	26	28.5
ΔI_{ph} (μ A) - 5 [#] W/O SVA	14.7	25.0	38.5	44.8	51.3
ΔI_{ph} (μ A) - 6 [#] With SVA	56.6	119.2	155.9	175.0	187.6

5. *There is insufficient direct evidence for the hypothesized mechanisms. I am more or less willing to accept the schematics in 4(a-c) are reasonable.*

Author replay:

Thanks for the reviewer's suggestion. We have supplemented the cross-sectional SEM imaging of different devices (Figure 4g-i). The features of these films were described by the schematic diagram shown in Figure 4a-c. For the spin-coated bilayer devices, when using the CHCl₃-soluble PC₆₁BM solution, the interface between PDPP3T and PC₆₁BM films was almost invisible (~ 184 nm, Figure 4g), indicating a high degree of mutual dissolution of these two films. In comparison, when the THF-soluble PC₆₁BM solution was used, the relatively blurred interface between the PDPP3T and the PC₆₁BM can be observed (Figure 4h), indicating the partial dissolution of these two films. Besides, an obvious interface was observed in the transferred bilayer devices (Figure 4i), suggesting no mutual dissolution between the first and the second layer. In words, combined the cross-sectional morphology (Figure 4g-i) with the film surface photographs (Figure 4d-f), it can be verified that the schematic diagram of bilayer films prepared with different methods is reasonable (Figure 4a-c). We supplemented it in the revised manuscript (see in page 12, first paragraph).

Figure 4. Schematic diagram of devices with Si/SiO₂/ OTS/ PDPP3T/ Au/ PC₆₁BM structure via different fabrication methods: (a) PC₆₁BM solution dissolved in chloroform solvent for spin-coating bilayer device (1[#] S-PDPP3T/ PC₆₁BM-CHCl₃), (b) PC₆₁BM solution dissolved in THF solvent for spin-coating bilayer device (2[#] S-PDPP3T/ PC₆₁BM-THF), (c) bilayer device via FTM (3[#] T-PDPP3T/ PC₆₁BM). The film surface photographs: (d) 1[#] S-PDPP3T/ PC₆₁BM-CHCl₃, (e) 2[#] S-PDPP3T/ PC₆₁BM-THF, (f) 3[#] T-PDPP3T/ PC₆₁BM. The cross-sectional SEM imaging of the films: (g) 1[#] S-PDPP3T/ PC₆₁BM-CHCl₃, (h) 2[#] S-PDPP3T/ PC₆₁BM-THF, (i) 3[#] T-PDPP3T/ PC₆₁BM.

6. However, I cannot accept that the process in Scheme 1 can be concluded, with this level of detail, from device measurements, Raman, UV-Vis, and AFM/SEM alone. The authors state “The effects of the diffusion interface layer on the device performance

are analyzed with an energy level diagram as shown in Scheme 1.” But as far as I can tell, this diagram was not measured, and is more or less entirely hypothesized.

Author replay:

Thanks for the reviewer’s suggestion. We have revised the state about the Scheme 1 as below:

The devices with SVA showed better performance than the devices without SVA treatment due to the presence of the diffusion interface layer. This phenomenon can be explained by the movement of the photogenerated carriers. For this purpose, the simplified physical model, which illustrates the movement of these carriers under two-electrodes conditions (without the gate electrodes, Scheme 1), was used in this work. It is worth mentioning that the effect of the gate voltages on the photocurrents will be discussed in the following session. In Scheme 1, the energy level diagram was provided to analyze the effects of the diffusion interface layer on the device performance enhancement. We supplemented it in the revised manuscript (see in page 20, first paragraph).

Furthermore, the energy levels of PDPP3T and PC₆₁BM were determined by ultraviolet photoelectron spectroscopy (UPS). The highest occupied molecular orbital (HOMO) levels were obtained from the valence band edges (Figure 2e) and the secondary electron cut-off (Figure 2f), which showed that the HOMO levels of the PDPP3T and PC₆₁BM were 5.24 and 5.96 eV, respectively (Table S1). The lowest unoccupied molecular orbital (LUMO) levels were determined by combing the optical bandgap from the UV-vis absorption spectra (Figure 2c and Figure S3) with the HOMO levels. The results show that the LUMO levels of the PDPP3T and PC₆₁BM were 3.89 and 4.1 eV, respectively (Table S1). We supplemented it in the revised manuscript (see in page 8, first paragraph).

Figure 2. Ultraviolet photoelectron spectrometer (UPS) spectra of valence band region (e) and secondary electron cut-off region (f).

Figure S3. The UV-vis absorption spectra of the PC₆₁BM film. Here, 1-0 absorption peak at 334 nm.

Scheme 1. Energy level diagram of the different device: (a) device without SVA and (c) device with SVA in the dark state, and (b) device without SVA and (d) device with SVA under illumination.

7. In particular, one of the biggest challenges with phototransistors is trying to disentangle the amplification (essentially photo-gating) vs charges being directly photogenerated and contributing to current (like in a solar cell). This is important because changes in threshold voltage are generally slow, but give rise to large EQE (often > 100%). It is clear that photogating / amplification is playing a role from the big shifts in threshold voltage (S10 (b) is a good example) but no distinction was made between this process and the current generation process hypothesized in Scheme 1.

Author replay:

Thanks for the reviewer's suggestion. As the reviewer stated, the biggest difference between phototransistors and solar cells is that the gate electrodes of phototransistors has a function of amplifying current. Based on the analysis of the carrier movement under two-electrodes conditions (Scheme 1), the effects of gate voltages were discussed to explore the photogating/amplification effects on the performance of phototransistors (Figure S20-21). When the phototransistors are in the turn-off state, the photocurrents usually increase linearly with the increase of light intensity due to a photoconductivity effect, which commonly exists in the photodiodes [Noh, Y.-Y. et al. *J. Appl. Phys.* **100**, 094501 (2006); Fuentes-Hernandez, C. et al. *Science* **370**, 698-701 (2020)]. In comparison, when the phototransistors are in the turn-on state, the photovoltaic effect is significant (equation 4) since the photovoltage is induced by the accumulation of trapped electrons [Noh, Y.-Y. et al. *J. Appl. Phys.* **100**, 094501 (2006)]. It can be seen that the photocurrents varied in the devices with SVA (6[#]) and without SVA treatment (5[#]). The photocurrents exhibited a non-linear relationship with the light intensity in turn-on state ($V_g = -30$ V, Figure S20-21), which satisfied equation (4). Therefore, the transconductance (G_M) values of the devices were calculated to estimate the photogating/amplification effect. Due to the formation of a diffusion interface in devices (6[#]) after the SVA treatment, the G_M value of devices (6[#]) was 5.3 μ S under weak light, lower than that in the devices without SVA (5[#]), in Figure 5a and Table 3. The smaller G_M value in devices (6[#]) could be attributed to that occurrence of a large drift in the threshold voltages under weak light (Figure 5a and Table 3). The G_M values of the SVA devices (6[#]) increased slowly from 5.3 to 6.7 μ S with the increase of light intensity. In contrast, the G_M value decreased from 7.3 to 4.3 μ S in devices without SVA (5[#]), which the opposite of the situation in the SVA devices (6[#]). According to Scheme 1, the diffusion interface of the SVA devices (6[#]) provided an additional donor/acceptor interface to separate photogenerated carriers. As a result, under strong light, the saturation condition of the photocurrents of the SVA devices (6[#]) is difficult to reach (Table S5), making the G_M value increase continuously (Figure S21b). The advantages of the diffusion interface layer were

highlighted by comparing the changes in G_M value of the two devices. We supplemented it in the revised manuscript (see in page 21, second paragraph).

Figure S20. (a) Dependence of photocurrent (I_{ph}) and threshold voltage drift (ΔV_{th}) on light intensity of the 5[#]-W/O SVA at $V_g = -30$ V, (b) dependence of G_M on light intensity of the 5[#]-W/O SVA at $V_g = -30$ V.

Figure S21. (a) Dependence of photocurrent (I_{ph}) and threshold voltage drift (ΔV_{th}) on light intensity of the 6[#]-W/O SVA at $V_g = -30$ V, (b) dependence of G_M on light intensity of the 6[#]-W/O SVA at $V_g = -30$ V.

Table S5. Performance parameters of devices with and without SVA treatment.

Light intensity (mW/cm ²)	0.004	0.02	0.13	1.52	4.5
ΔV_{th} (V) - 5 [#] W/O SVA	2	5	8	10	12
ΔV_{th} (V) - 6 [#] With SVA	10.5	19.5	24	26	28.5
ΔI_{ph} (μ A) - 5 [#] W/O SVA	14.7	25.0	38.5	44.8	51.3
ΔI_{ph} (μ A) - 6 [#] With SVA	56.6	119.2	155.9	175.0	187.6

8. I also have a few more minor comments:

(1) In the introduction these two statements need to be elaborated upon or provided with a reference:

- “However, the acceptor phase in traditional single layer bulk heterojunction (BHJ) photodetector needs a high concentration, which easily causes electron/hole recombination.”
- “It is reported that bilayer devices (channel transport layer/ BHJ layer) can effectively reduce the effects of acceptors on device performance”

Author replay:

Thanks for the reviewer’s suggestion. Relevant literatures have been added, such as: “Nevertheless, the acceptor phase in traditional single layer bulk heterojunction (BHJ) photodetector needs a high concentration, which easily causes electron/hole recombination [Han, H. et al. ACS Appl. Mater. Inter. **9**, 628-635 (2017)].”

“It has been reported that the design of bilayer devices (channel transport layer/ BHJ layer) could effectively enhance devices performance [Gao, Y. et al. Adv. Mater. **31**, 1900763 (2019)].”

(2) Figure 2(b) needs a lateral scale bar.

Author replay:

Thanks for the reviewer's suggestion. We supplemented the scalebar of AFM diagram in Figure 2b, and its scanning range is $2 \times 2 \mu\text{m}^2$ (see in page 9, Figure notes in Figure 2).

Figure 2. The TEM (a), AFM (b) of different films.

(3) The peaks in Figure 2(d) need to be labeled in the caption. The caption to Figure 2(d) also should explain what is being shown since this is not what one expects when seeing a “GIWAXS image”. I assume this is a line cut. The text mentions a different abbreviation “GIXRD” to “GIWAXS” in the caption. In either case, the abbreviation needs to be spelled out on first use.

Author replay:

Thanks for the reviewer's suggestion. We had a wrong mark in the Figure notes (Figure 2), where “GIWAXS” should change into “grazing-incidence X-ray diffraction (GIXRD) patterns”.

Figure 2. The grazing-incidence X-ray diffraction (GIXRD) patterns (d) of different films.

(4) The labels “0-0”, “0-1” and “1-0” in 3(c) need to be explained in the caption.

Author replay:

Thanks for the reviewer’s suggestion. The 0-0 absorption peak at 850 nm, 0-1 absorption peak at 766 nm, 1-0 absorption peak at 334 nm in Figure 3c.

Figure 3. (c) the absorption spectra of the transferred films after removing films above the PEDOT:PSS layer with different SVA time.

(5) The section referring to Raman needs to be re-written to be clearer. It was very difficult to follow in its current form.

Author replay:

Thanks for the reviewer's suggestion. We have added the relevant descriptions in the Figure notes of the Figure 3 and Figure S2. For example, the PDPP3T-25 °C and PDPP3T:PC₆₁BM-25 °C represent the Raman test of PDPP3T and PDPP3T:PC₆₁BM film at 25 °C in Figure 3a; the PDPP3T-25 °C and PC₆₁BM-25 °C represent the Raman test of PDPP3T and PC₆₁BM film at 25 °C in Figure S2a.

At the same time, we have added the detailed Raman test process in the experimental section: the Si/ SiO₂/ OTS was used as the substrate, the active layer was selected PDPP3T, PC₆₁BM, PDPP3T:PC₆₁BM, PDPP3T/PC₆₁BM and PDPP3T/PDPP3T:PC₆₁BM to prepare related samples. The Raman test data at the same depth were selected to achieve in-situ Raman characterization under different annealing temperature. Besides, PDPP3T-25 °C, PC₆₁BM-25 °C and PDPP3T:PC₆₁BM-25 °C were tested at 25°C, the PDPP3T/PC₆₁BM and PDPP3T/PDPP3T:PC₆₁BM films were tested under annealing conditions of 25-130 °C and 25-100 °C, respectively. The variation of the characteristic Raman peak of PC₆₁BM at 1460 cm⁻¹ was chosen to analyze the effect of annealing on PC₆₁BM (see in page 25, second paragraph).

Figure 3. (a) In situ Raman spectra of PDPP3T/PDPP3T:PC₆₁BM bilayers with SVA at different temperatures, (b) dependence of normalized PC₆₁BM Raman peak in PDPP3T/PDPP3T:PC₆₁BM layer on annealing temperature (data comes from Figure 3a). Here, the PDPP3T-25 °C and PDPP3T:PC₆₁BM-25 °C represent the Raman test of PDPP3T and PDPP3T:PC₆₁BM film at 25 °C in Figure 3a.

Figure S2. In situ Raman spectra of PDPP3T/PC₆₁BM bilayers with solvent vapor annealing (SVA) treatment at different temperatures, (b) dependence of normalized PC₆₁BM Raman peak in PDPP3T/PC₆₁BM layer on annealing temperature (data comes from Figure S2a). Here, the PDPP3T-25 °C and PC₆₁BM-25 °C represent the Raman test of PDPP3T and PC₆₁BM film at 25 °C in Figure S2a.

(6) Why was turn on voltage ($V_{\{0\}}$) used rather than the more conventional threshold voltage ($V_{\{Th\}}$)?

Author replay:

Thanks for the reviewer's suggestion. This is because by using turn on voltage on the transfer curve, the impact of light on device performance can be analyzed more clearly and intuitively [Noh, Y.-Y. et al. *J. Appl. Phys.* **100**, 094501 (2006); Shou, M. et al. *Adv. Opt. Mater.* **9**, 2002031 (2021)].

(7) I don't agree with the conclusion that "The hysteresis window of the FTM-based bilayer device 6[#] obtained after SVA reduced significantly, indicating that the defects/traps in the film could be effectively eliminated..." hysteresis is not the only possible manifestation of trap states.

Author replay:

Thanks for the reviewer's suggestion. The existence of defects/traps causes the capture and release process of carriers to occur during the forward scanning and backward scanning of the transfer curve, forming a hysteresis window [Cho, B. et al. *Adv. Funct. Mater.* **21**, 2806-2829 (2011)]. Of course, as the reviewer stated, hysteresis is not the only possible manifestation of trap states. Hence, the sentence "The hysteresis window of the FTM-based bilayer device 6[#] obtained after SVA reduced significantly, indicating that the defects/traps in the film could be effectively eliminated..." should change into "The hysteresis window of the FTM-based bilayer SVA devices (6[#]) reduced significantly, indicating that the defects/traps in the films could be effectively eliminated" (see in page 16, first paragraph).

(8) Be aware that intrinsic has a very specific meaning in semiconductors and using the phrase "intrinsic electrons" so frequently may be confusing to the general reader.

Author replay:

Thanks for the reviewer's suggestion. We have changed the "intrinsic electrons" into the "electrons".

Peer review comments, second round review–

Reviewer #1 (Remarks to the Author):

Dear Mr. Han, dear Mr. Zhang,

thank you for very thoroughly revising your manuscript and satisfactorily addressing every point in my review.

I'd still recommend a revision with respect to language before publication. This would make your manuscript easier to read and may increase the impact your work can have in the research community.

Scientifically I find the manuscript to be in much better shape now.

Best regards,
Felix Dollinger

Reviewer #2 (Remarks to the Author):

I thank the authors for the extensive revisions they have made to the manuscript and the additional data they have provided. From the point of view of scientific soundness, I am happy to recommend this manuscript for publication in its current form.

Manuscript ID: *NCOMMS-21-31071A-Z*

Title: **Diffusion Interface Layer Controlling the Acceptor Phase for Ultrahigh Photosensitivity Bilayer Near-Infrared Polymer Phototransistors**

Journals: **Nature Communications**

Reviewers' comments:

Reviewer #1 (Remarks to the Author):

Dear Mr. Han, dear Mr. Zhang,

Thank you for very thoroughly revising your manuscript and satisfactorily addressing every point in my review.

I'd still recommend a revision with respect to language before publication. This would make your manuscript easier to read and may increase the impact your work can have in the research community.

Scientifically I find the manuscript to be in much better shape now.

Author replay:

Thanks for the reviewer's suggestion. We asked our colleagues to review our manuscript and polish the English language. The misleading expressions, pointed out by the reviewers, have been changed into clearer ones. A version with tracked changes in the file named "Manuscript (a version with tracked changes)" has been supplied for you to check the revision easily.

For example:

- (a) the original sentence "the dark currents of the photodiodes are larger than that of the phototransistor since the electrode distance (< 500 nm) of the photodiodes is much shorter (phototransistors: > 5 μm)" changes into "the dark currents of the

photodiodes are larger than that of the phototransistor due to the short electrode distance (photodiodes: < 500 nm, phototransistors: > 5 μm)” (see in page 2, the last paragraph).

- (b) the original sentence “to further improve and control the performance of bilayer NIR photodetectors, the uneven distribution of the acceptor phases caused by the interfacial mutual dissolution was found to be the primary technical challenge” changes into “in order to further improve and control the performance of bilayer NIR photodetectors, it was found that the uneven distribution of the acceptor phases caused by the interfacial mutual dissolution was a major technical challenge” (see in page 4, the first paragraph).
- (c) the original sentence “the PEDOT:PSS layer was used to assist the measurement of the changes in the PC₆₁BM absorption peaks” changes into “the PEDOT:PSS layer assists in measuring the changes of absorption peaks of PC₆₁BM” (see in page 9, the last paragraph).
- (d) the original sentence “These results further indicate that the acceptor phases of device 4[#] diffused into the channel transport layer to form a diffusion interface after SVA treatment” changes into “These results further indicate that the acceptor phases of device 4[#] diffused into the channel transport layer after SVA treatment, forming a diffusion interface” (see in page 14, the second paragraph).
- (e) the original sentence “The recombination of electrons and holes during the operation of the devices results in a big hysteresis window, high off-state dark currents, and turn-on voltages, which have an adverse effect on the performance of the devices” changes into “In the operation process of devices, the recombination of electrons and holes will result in a big hysteresis window, high off-state dark currents, and turn-on voltages, which will adversely affect devices performance” (see in page 21, the first paragraph).

Reviewer #2 (Remarks to the Author):

I thank the authors for the extensive revisions they have made to the manuscript and the additional data they have provided. From the point of view of scientific soundness, I am happy to recommend this manuscript for publication in its current form.

Author replay:

Thanks for the reviewer's suggestion. Your suggestion greatly improved the quality of our manuscript.